# FROM GENERALIZATION ANALYSIS TO OPTIMIZATION DESIGNS FOR STATE SPACE MODELS

## ABSTRACT

A State Space Model (SSM) is a foundation model in time series analysis, which has recently been shown as an alternative to transformers in sequence modeling. In this paper, we theoretically study the generalization of SSMs and propose improvements to training algorithms based on the generalization results. Specifically, we give a *data-dependent* generalization bound for SSMs, showing an interplay between the SSM parameters and the temporal dependencies of the training sequences. Leveraging the generalization bound, we (1) set up a scaling rule for model initialization based on the proposed generalization measure, which significantly improves the robustness of the output value scales on SSMs to different temporal patterns in the sequence data; (2) introduce a new regularization method for training SSMs to enhance the generalization performance. Numerical results are conducted to validate our results.

## 1 INTRODUCTION

Sequence modeling has been a long-standing research topic in many machine learning areas, such as speech recognition (Hinton et al., 2012), time series prediction (Li et al., 2019), and natural language processing (Devlin et al., 2019). Various machine learning models have been successfully applied in sequence modeling to handle different types of sequence data, ranging from the (probabilistic) Hidden Markov model (Baum & Petrie, 1966) to deep learning models, e.g., Recurrent Neural Networks (RNNs), Long Short-Term Memory units (Hochreiter & Schmidhuber, 1997), Gated Recurrent Unit (Chung et al., 2014), and transformers (Vaswani et al., 2017). In this paper, we focus on the state space model (SSM), which has a simple mathematical expression[1]: $h'(t) = Ah(t) + Bx(t), y(t) = Ch(t) + Dx(t)$ where $h(t)$ is the hidden state, $x(t)$ is the input sequence, $y(t)$ is the output sequence and $A, B, C, D$ are trainable parameters. Recent studies have demonstrated the power of SSMs in deep learning. For example, it was shown in Gu et al. (2022a) that by a new parameterization and a carefully chosen initialization, the structured state space sequence (S4) model achieved strong empirical results on image and language tasks. Following the S4 model, more variants of SSMs are proposed, e.g., the diagonal SSM (Gu et al., 2022b; Gupta et al., 2022) the S5 model (Smith et al., 2023), the H3 model (Fu et al., 2023), the GSS model (Mehta et al., 2023), and the Hyena Hierarchy (Poli et al., 2023).

Theoretical analysis and understanding of the approximation and optimization of SSMs are well studied in the literature such as (Li et al., 2021; 2022; Gu et al., 2022a; 2023). Since the SSM can be regarded as a continuous linear RNN model (Li et al., 2022), most generalization analysis of SSMs is based on the generalization theory of RNNs (Zhang et al., 2018; Chen et al., 2019; Tu et al., 2019). However, these previous works did not study the effects of the temporal dependencies in the sequence data on the SSM generalization (See more details on the comparison in Section 4.1). As an attempt to understand the relationship between the temporal dependencies and the generalization performance, this paper aims to provide a generalization bound that connects the memory structure of the model with the temporal structure of the data. We can, in turn, use the proposed bound to guide us in designing new algorithms to improve optimization and generalization. Specifically, we discover two roles for the proposed generalization measure: (1) generalization bound as an *initialization scheme*; (2) generalization bound as a *regularization method*. The common initialization method for the S4 model and its variants follows from the HiPPO framework (Gu et al., 2022a;

---

[1]To simplify the analysis, we omit the skip connection by letting $D = 0$

2023), which is based on the prerequisite that the training sequence data is stable. To improve the robustness of the output value scales on SSMs to different temporal patterns in the sequence data, we consider to rescale the initialization of SSMs with respect to the generalization measure. This new initialization scheme makes the SSMs more resilient on their initial output value scales to variations in the temporal patterns of the training data. Except for the initialization setup, our generalization bound can also be served as a regularizer. Regularization methods like weight decay and dropout are widely applied to training SSMs, but the hidden state matrix $A$ is not regularized because its imaginary part controls the oscillating frequencies of the basis function $e^{At}B$ (Gu et al., 2022b). By taking into account the interaction between the SSM structure and the temporal dependencies, we introduce a new regularization method based on our bound, and it can be applied to the hidden state space to improve the generalization performance. When combining the initialization scheme and the regularization method, our method is applicable to various tasks, ranging from image classification to language processing, while only introducing a minimal computational overhead. To summarize, our contributions are as follows:

- We provide a data-dependent generalization bound for SSMs by taking into account the temporal structure. Specifically, the generalization bound correlates with the memory structure of the model and the (auto)covariance process of the data. It indicates that instead of the weight or the data norm, it is the interplay between the memory structure and the temporal structure of the sequence data that influences the generalization.

- Based on the proposed generalization bound, we setup an initialization scaling rule by adjusting the magnitude of the model parameters with respect to the generalization measure at initialization. This scaling rule improves the robustness of the initial output value scales on SSMs across different temporal patterns of the sequence data.

- Apart from the initialization scheme, we design a new regularizer for SSMs. Unlike weight decay, our regularizer does not penalize the parameter norm but encourages the model to find a minimizer with lower generalization bound to improve the generalization performance.

## 2 RELATED WORKS

Since a SSM is also a continuous linear RNN, there are three lines of research that are related to our work: generalization of RNNs, temporal structure analysis on RNNs, and optimization of SSMs.

**Generalization of RNNs.** Existing works on the generalization of RNNs focus on the generalization error bound analysis. Specifically, in the early two works of Dasgupta & Sontag (1995) and Koiran & Sontag (1998), VC dimension-based generalization bounds were provided to show the learnability of RNNs. In recent studies, Zhang et al. (2018); Chen et al. (2019); Tu et al. (2019) proved norm-based generalization bounds, improving the VC dimension-based bounds by the Rademacher complexity technique (Bartlett & Mendelson, 2002) under the uniform-convergence framework. In the overparameterization settings, it was shown in Allen-Zhu & Li (2019) that RNNs can learn some concept class in polynomial time given that the model size is large enough. These generalization bounds, however, do not take into account the temporal dependencies and their effects on generalization. In this work, we provide a new generalization bound by combining the memory structure of the model and the temporal structure of the data.

**Temporal structure analysis on RNNs.** Sequence data has long-range temporal dependencies across the time domain, which notably set it apart from non-sequence data. Recent studies have studied the effects of such temporal dependencies on the approximation and optimization of RNNs. For example, in the two works of Li et al. (2021; 2022), a "curse of memory" phenomenon was discovered when using linear RNNs to model the temporal input-output relationships. Particularly, when the target relationship between the input and output has a long-term memory, then both approximation and optimization become extremely challenging. In Wang et al. (2023), the "curse of memory" phenomenon on approximation and optimization was extended to non-linear RNNs based on the temporal relationships. In this paper, we conduct a fine-grained analysis on the effects of the temporal structure analysis on the *generalization* of RNNs.

**Optimization of SSMs.** RNN optimization is known for two issues: training stability and computational cost (Bengio et al., 1994; Pascanu et al., 2013). To address these two issues and capture the long dependencies more efficiently in sequence modeling, the S4 model was proposed by in-

troducing new paraemterization, initialization and discretization (Gu et al., 2022a). Recent variants for the S4 model simplified the hidden state matrix by a diagonal matrix to enhance computational efficiency (Gu et al., 2022b; Gupta et al., 2022; Smith et al., 2023; Orvieto et al., 2023). Regularization methods are also applied for SSMs to prevent overfitting, such as dropout, weight decay and the data continuity regularizer (Qu et al., 2023). However, the principled way to regularize and initialize the parameters still remains to be explored. In this study, we design a new regularization and initialization scheme to improve both optimization and generalization.

## 3 PRELIMINARIES

In this section, we briefly introduce the SSM in Section 3.1 and the motivation for optimization designs based on the generalization analysis in Section 3.2.

### 3.1 INTRODUCTION TO SSMs

In this paper, we consider the following single-input single-output SSM,

$$h'(t) = Ah(t) + Bx(t), \quad y(t) = Ch(t), \quad t \geq 0 \tag{1}$$

where $x$ is the input from an input space[2] $\mathcal{X} := C_0(\mathbb{R}_{\geq 0}, \mathbb{R})$; $y(t) \in \mathbb{R}$ is the output at time $t$; $h(t) \in \mathbb{R}^m$ is the hidden state with $h(0) = 0$; $A \in \mathbb{R}^{m \times m}, B \in \mathbb{R}^{m \times 1}, C \in \mathbb{R}^{1 \times m}$ are trainable parameters. Then (1) has an explicit solution $y(t) = \int_0^t \rho_\theta(s) x(t-s) ds$, where $\rho_\theta(s) := Ce^{As}B$ with $\theta = (C, A, B)$. The function $\rho_\theta(s)$ captures the memory structure of the model and the temporal input-output relationship (Li et al., 2022). For the S4 model and its variants (Gu et al., 2022a;b; Gupta et al., 2022; Gu et al., 2023), (1) is usually discretized by the Zero-Order Hold method, i.e., given a timescale $\Delta \in \mathbb{R}$, $h_{k+1} = \bar{A}h_k + \bar{B}x_k, \quad y_k = \bar{C}h_k, \quad k = 0, 1, \ldots$, where $\bar{A} = e^{\Delta \cdot A}, \bar{B} = (\bar{A} - \mathbb{I}_m)A^{-1}B, \bar{C} = C$. Then, $y_k = \bar{C}\bar{A}^k\bar{B}x_0 + \bar{C}\bar{A}^{k-1}\bar{B}x_1 + \ldots + \bar{C}\bar{B}x_k = [\bar{K} * x]_k$ where $\bar{K} = (\bar{C}\bar{B}, \bar{C}\bar{A}\bar{B}, \ldots, \bar{C}\bar{A}^k\bar{B})$ and $*$ represents to convolution.

### 3.2 MOTIVATION: A LINEAR REGRESSION MODEL

In this subsection, we use a linear regression model on non-sequential data as an example to illustrate the connection between the generalization analysis and the optimization designs. This example then motivates us to extend the connection to SSMs on sequential data.

**Linear regression.** We consider a simple linear model $y = \theta^\top x$ with input $x \in \mathbb{R}^d$, output $y \in \mathbb{R}$ and parameter $\theta \in \mathbb{R}^d$. Let the training data $\{(x_i, y_i)\}_{i=1}^n$ be i.i.d. sampled from a distribution $\mathcal{D}$ such that $\|x_i\|_2 = r, |y_i| \leq 1 (\forall i \in [1:n])$. Define the empirical risk $\mathcal{L}_n(\theta) := \frac{1}{n}\sum_{i=1}^n (\theta^\top x_i - y_i)^2$ and the population risk $\mathcal{L}_\mathcal{D}(\theta) := \mathbb{E}_{x,y}[(\theta^\top x - y)^2]$. Then given a norm-constrained space $\Theta := \{\theta \in \mathbb{R}^d : \|\theta\|_2 \leq R\}$, with probability at least $1 - \delta$ over $\mathcal{D}$,

$$\sup_{\theta \in \Theta} |\mathcal{L}_n(\theta) - \mathcal{L}_\mathcal{D}(\theta)| \leq (rR + 1)^2 \cdot \mathcal{O}(\sqrt{\log(1/\delta)/n}). \tag{2}$$

This is a well-known norm-based generalization bound based on the Rademacher theory (Mohri et al., 2012), and we provide a proof in Appendix B for completeness. Notice that the key term $r^2R^2$ in the generalization bound (2) is also an upper bound for the magnitude of the linear model output, i.e., $\sup_{\theta \in \Theta}(\theta^\top x_i)^2 \leq r^2R^2$. Thus, we connect the model stability with the generalization bound stability, and this connection induces an initialization scheme for the initialization $\theta^{(0)}$ by setting $\|\theta^{(0)}\|_2 \sim \mathcal{O}(1/r)$. In particular, if we normalize each input $x_i$ such that $r$ is also $\mathcal{O}(1)$, then $\|\theta^{(0)}\|_2 \sim \mathcal{O}(1)$. Since $\theta^{(0)} \in \mathbb{R}^d$, one possible initialization scheme is that $\theta^{(0)}$ follows a Uniform distribution $U[-1/\sqrt{d}, 1/\sqrt{d}]$, which corresponds to the Kaiming initialization (up to some constant) (He et al., 2015). When treating the term $r^2R^2$ as a regularizer to improve the generalization, we get the weight decay method, i.e., the $\ell_2$ regularization w.r.t. $\|\theta\|_2^2$. We summarize the above logic chain that connects the generalization analysis with optimization designs in Figure 1. Now for SSMs, we extend the generalization analysis from non-sequential data to sequential data by taking into account the temporal structure of the data. This linear regression example motivates us to apply the same logic diagram (Figure 1) to the SSMs, and this is exactly what we are going to present in the following part of this paper.

---

[2]A linear space of continuous functions from $\mathbb{R}_{\geq 0}$ to $\mathbb{R}$ that vanishes at infinity.

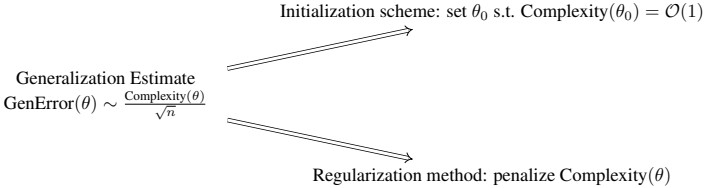

Figure 1: The logic diagram goes from generalization analysis to optimization designs.

## 4 MAIN RESULTS

In this section, we first give a generalization bound for SSMs in Section 4.1, then we design a new initialization scheme in Section 4.2 based on this proposed bound. Apart from the initialization scheme, we introduce a new regularization method in Section 4.3. Finally, we conduct experiments to test the initialization scheme and the regularization method in Section 4.4.

### 4.1 A GENERALIZATION BOUND OF SSMs

In this section, we present a generalization bound for the SSM (1) and reveal the effects of the temporal dependencies on the generalization performance. We show that our bound gives a tighter estimate compared with previous norm-based bounds through a toy example. Following the same notation in Section 3.1, we define the empirical risk $R_n(\theta)$ and the population risk $R_x(\theta)$ as

$$R_n(\theta) := \frac{1}{n} \sum_{i=1}^{n} \left| \int_0^T \rho_\theta(T-s)x_i(s)ds - y_i \right|^2, \quad R_x(\theta) := \mathbb{E}_x \left| \int_0^T \rho_\theta(T-s)x(s)ds - y \right|^2$$

where $T > 0$ is some finite terminal time, the training sequence data $\{x_i(t)\}_{i=1}^n$ are independently sampled from a stochastic process with mean $\mathbb{E}[x(t)] := \mu(t)$ and covariance $\mathbb{E}[(x(s)-\mu(s))(x(t)-\mu(t))] := K(s,t)$, and the label $y$ is generated by some underlying functional $H_T : \mathcal{X} \to \mathbb{R}$, i.e., $y = H_T(x)$. We assume that $|y| \le 1$ for any $x \in \mathcal{X}$, otherwise, we truncate the value of the label to 1. In the next, we make an assumption on the normalized process $\tilde{x}(t) := (x(t) - \mu(t))/\sqrt{K(t,t)}$:

**Assumption 1.** *The normalized process $\tilde{x}(t)$ is (1): almost surely Hölder continuous, i.e., $\exists L, H > 0, s.t. \forall s, t \in [0,T], |\tilde{x}(s) - \tilde{x}(t)| \le L|s-t|^H a.s.;$ (2): is $\sigma^2$-sub-Gaussian for every $t \in [0,T]$, i.e., $\exists \sigma > 0, s.t. \forall u > 0, P(|\tilde{x}(t)| \ge u) \le 2\exp(-u^2/2\sigma^2)$ for any $t \in [0,T]$.*

We leave the discussion of the assumption after the statement of the main theorem. Now we proceed to bound generalization gap $|R_x(\theta) - R_n(\theta)|$ by establishing uniform convergence of the empirical risk to its corresponding population risk, as stated in following theorem:

**Theorem 1.** *For a SSM $\int_0^T \rho_\theta(T-s)x(s)ds$, following notations and settings in Section 3.1 & 4.1, we define $\psi(\Theta) := \sup_{\theta \in \Theta} \int_0^T |\rho_\theta(T-s)| \sqrt{K(s,s)}ds + \sup_{\theta \in \Theta} \left| \int_0^T \rho_\theta(T-s)\mu(s)ds \right|$. Then under Assumption 1, given a parameter space $\Theta$ for $\theta$, for any $\delta \in (0,1)$, with probability at least $1 - \delta$ over the training sequences,*

$$\sup_{\theta \in \Theta} |R_x(\theta) - R_n(\theta)| \le (\psi(\Theta) + 1)^2 \cdot \mathcal{O}(\log^{3/2}(Tn/\delta)/\sqrt{n}). \tag{3}$$

Where $\mathcal{O}$ hides a constant that depends on $\sigma, L, H$. The proof is given in Appendix E. We see that this bound decreases to zero as the sample size $n \to \infty$, provided that the terminal time $T$ is finite and the supremum term in (3) is bounded. Theorem 1 captures the temporal dependencies of the sequence data on the SSM generalization, yielding that the mean and variance at each length position together play important roles in generalization analysis. Specifically, as long as $\psi(\Theta)$ is small, the generalization gap is small. Since the function $\rho_\theta(s)$ is exponentially decay, we do not require the mean and variance to be uniformly small along the time $t$ to get a small generalization gap.

**Proof sketch.** The proof is based on Rademacher theory (Bartlett & Mendelson, 2002). The main difficulty is to bound the Rademacher complexity of the SSM function $\int_0^T \rho_\theta(T-s)x(s)ds$ for a stochastic process $x(s)$. We first use the Hölder inequality to get an upper bound for the Rademacher complexity w.r.t. the normalized process $\tilde{x}(s)$, then combining Hölder continuity and the heavy-tail property in Assumption 1, we show the finiteness of $\sup_{s \in [0,T]} \tilde{x}(s)$. Finally we use an $\varepsilon$-net argument to give an explicit bound for the Rademacher complexity, which then finishes the proof.

**Discussions of Assumption 1.** This assumption contains two parts. Hölder continuity is used to bound $\sup_{s \in [0,T]} \tilde{x}(s)$ and the Rademacher complexity of the SSM function class. By the Kolmogorov continuity theorem (Stroock & Varadhan, 1997), Hölder continuity covers a wide range of random process that satisfies certain inequalities for its moments. For the sub-Gaussian property, it ensures $\tilde{x}(s)$ is bounded in a finite time set with high probability. Sub-Gaussian random variables include Gaussian and any bounded variables. Specifically, for image classification tasks with flattened image pixels, if the range of the pixel values is a finite class, then the Hölder continuity condition can be dropped. We leave more detailed discussions and provide some concrete examples that satisfy Assumption 1 in Appendix C.

**Comparison to previous bounds.** Since a SSM is also a continuous linear RNN model, we compare (3) with previous bounds for linear RNNs. In Chen et al. (2019), a generalization bound $\widetilde{\mathcal{O}}\left(\|x\|_2 \|B\|_2 \|C\|_2 \|A\|_2 / \sqrt{n}\right)$ is provided, where $\|x\|_2$ is the 2-norm of the discrete input sequence. In the continuous case, $\|x\|_2$ corresponds to the $L^2$ norm w.r.t. a Dirac measure. By changing the matrix 2-norm to matrix 1-norm, Tu et al. (2019) shows another similar generalization bound. These bounds separate the data complexity and the model complexity by the data norm and the model parameter norm individually, and do not account for the temporal dependencies across the time domain. In this work, instead, we incorporate the temporal dependencies via the sequence statistics (mean and variance) to get a generalization bound. Next, we use a toy example to illustrate that our bound gives a tighter estimation. Given a stochastic process $\{x(t)\}_{t \in [0,T]}$ with mean $\mu(t)$ and covariance $K(s,t)$, we consider the following two upscale transformations (by increasing $T$ to $2T$):

1. left zero padding: $x_1(t) = 0, \ t \in [0, T); \quad x_1(t) = x(t - T), \ t \in [T, 2T]$
2. right zero padding: $x_2(t) = x(t), \ t \in [0, T]; \quad x_2(t) = 0, \ t \in (T, 2T]$

Then the two SSM outputs are given by $y_i(2T) = \int_0^{2T} \rho_\theta(2T - s)x_i(s)ds$ for $i = 1, 2$. Hence,

$$y_1(2T) = C \int_0^T e^{A(T-s)} Bx(s)ds, \quad y_2(2T) = Ce^{AT} \int_0^T e^{A(T-s)} Bx(s)ds. \tag{4}$$

We see that the magnitude of $y_1(2T)$ and $y_2(2T)$ differs with an exponential factor $e^{AT}$. Since all the eigenvalues of $A$ have negative real part, $y_2(2T) \to 0$ as $T$ increases. Hence, the right zero padding transformation degenerates the SSM function class to a zero function class for large $T$, inducing a *minimal* generalization gap that only contains the statistical sampling error (see (3) by letting $K(s,s) = \mu(s) = 0$). Therefore, a desired generalization bound should reflect such a difference caused by the different temporal dependencies. However, previous norm-based generalization bounds do not capture such a difference for these two transformations as they produce the same $L^2$ norm for the input sequence. Let us see what happens for our proposed generalization measure. For the left zero padding, the key term in (3) becomes

$$\int_0^T \left| Ce^{A(T-s)} B \right| \sqrt{K(s,s)} ds + \left| \int_0^T Ce^{A(T-s)} B\mu(s)ds \right| + 1 \tag{5}$$

For the right zero padding, the key term in (3) becomes

$$\int_0^T \left| Ce^{AT} e^{A(T-s)} B \right| \sqrt{K(s,s)} ds + \left| \int_0^T Ce^{AT} e^{A(T-s)} B\mu(s)ds \right| + 1 \tag{6}$$

The detailed derivations are given in Appendix D. By the same argument, our bound (3) indeed captures the difference on the magnitude of the generalization performance for these two sequence transformations. In particular, as $T \to \infty$, (6) reduces to 1, which yields a minimal generalization gap as expected for the zero function class. In that sense, we get a tighter bound for the SSMs.

**Zero shot transferability.** A benefit of SSMs is the zero-shot transfer to other sampling frequencies (i.e., the timescale measure in continuous case). For example, for a SSM function $y_T = \int_0^T \rho_\theta(T - s)x(s)ds$, if we downscale the input sequence $x(s)$ by half of the sampling frequency, then the SSM output becomes $y_T = \int_0^{T/2} \rho_\theta(T - 2s)x(2s)ds$, which equals to $\int_0^T \frac{1}{2}\rho_\theta(T - s)x(s)ds$. Now for a new SSM parameter $\tilde{\theta} = (2C, A, B)$, we have $\rho_{\tilde{\theta}}(s) = 2\rho_\theta(s)$, indicating that by simply modifying the SSM parameters, one can transfer the model to half the sampling frequency while

keeping the output invariant. One advantage for our generalization measure is that it is also zero shot transferable. To see this, we use the same example here. Under the downscale sampling, both $\int_0^T |\rho_\theta(T-s)| \sqrt{K(s,s)}ds$ and $\left|\int_0^T \rho_\theta(T-s)\mu(s)ds\right|$ remain invariant for the new parameter $\tilde{\theta}$ because $\sqrt{K(s,s)}$ and $\mu(s)$ have the same scaling as $x(s)$. Similarly, other sampling frequencies are also zero shot transferable for our generalization measure by simply adjusting the SSM parameters.

## 4.2 GENERALIZATION BOUND AS AN INITIALIZATION SCHEME

In this section, we design a scaling rule for the SSM parameters at initialization baed on the generalization bound (3). This new initialization scheme improves the robustness of the initial output value scales on SSMs across different temporal patterns of the sequence data.

Our proposed initialization scheme is built on the HiPPO based initialization (Gu et al., 2023), which is a *data independent* initialization method. Specifically, the HiPPO framework initializes the hidden state matrices $A, B$ to produce orthogonal basis functions, and the matrix $C$ to be standard normal for training stability. However, the argument for the training stability relies on the prerequisite that the input sequence is constant along the length (Gu et al. (2023, Corollary 3.4)), which is restricted as the long-range dependencies may lead to very different temporal patterns on the input sequence. As the dashed lines in the left and the right part of Figure 2 show, the SSM output value scale and the loss value scale under the HiPPO based initialization vary much across different temporal dependencies, making the loss values inconsistent during training. To address this issue, we follow the logic diagram in Figure 1 by adjusting the generalization complexity to be $\mathcal{O}(1)$. Specifically, we extract the dominant term in the generalization bound (3):

$$\tau(\theta) := \left(\int_0^T |\rho_\theta(T-s)| \sqrt{K(s,s)}ds + \left|\int_0^T \rho_\theta(T-s)\mu(s)ds\right|\right)^2. \tag{7}$$

Notice that $\rho_\theta(s) = Ce^{As}B$, if we rescale $C$ to $\xi C$ for some $\xi \in \mathbb{R}$, we have $\tau(\tilde{\theta}) = \xi^2 \cdot \tau(\theta)$ for $\tilde{\theta} = (\xi C, A, B)$. This induces a new initialization scheme, i.e., once the parameters $\theta = (C, A, B)$ are initialized by the HiPPO method, we rescale $C$ to $\tilde{C}$ such that

$$\tilde{C} = \frac{1}{\sqrt{\tau(\theta)}} \cdot C = \frac{1}{\int_0^T |\rho_\theta(T-s)| \sqrt{K(s,s)}ds + \left|\int_0^T \rho_\theta(T-s)\mu(s)ds\right|} \cdot C \tag{8}$$

This rescaling method guarantees the SSM output value to be bounded at initialization for *any* stochastic process that satisfies Assumption 1, ensuring the robustness of the initial loss value scales on SSMs across different temporal dependencies. We formalize the statement in Proposition 1.

**Proposition 1.** *Consider a SSM $\int_0^T \rho_\theta(T-s)x(s)ds$ with $\theta = (C, A, B)$, for any stochastic process $x(s)$ that satisfies Assumption 1, let $\tilde{C}$ given by the rescaling method (8), then for $\tilde{\theta} := (\tilde{C}, A, B)$, we have $\mathbb{E}_x \left[\left|\int_0^T \rho_{\tilde{\theta}}(T-s)x(s)ds\right|\right] \leq \mathcal{O}(\sqrt{\log T})$.*

The proof is provided in Appendix F. Proposition 1 shows that the SSM output values are uniformly bounded over all the stochastic processes that satisfy Assumption 1, even when the input sequence is not almost surely bounded. This improves the robustness of the output value scales on SSMs in the sense that the scale of the output value does not depend on the variations of the temporal structures. It is worth noting that different from the data normalization methods such as min-max normalization and standardization, our rescaling method only changes the model parameters. This is important because normalization on the data numerical values in language tasks can lead to loss of crucial information. For example, mathematical expressions like "$\max(1, 9) = 9$" have a contextual meaning where normalizing could result in the loss of structured information essential to understand.

**Implementation.** In the practical training, the SSMs used for tasks such as image classification or language processing are usually deep and high dimensional ($d > 1$), while our initialization scheme (8) is designed based on the one-dimensional shallow SSM. To extend to high-dimensional SSMs, we empirically treat all features to be independent and calculate $\tau(\theta)$ by its average along the feature dimension. for a $k$-layer SSM with the initial matrix $C_1, \ldots, C_k$ at each layer, we first calculate the complexity measure $\tau_1(\theta)$ for the first layer and rescale $C_1$ by $C_1/\sqrt{\tau_1(\theta)}$. Then we calculate

the complexity measure $\tau_2(\theta)$ for the second layer for the updated input sequence of layer 2 and rescale $C_2$ by $C_2/\sqrt{\tau_2(\theta)}$. We repeat this process until the last layer. We describe the complete procedures for one-layer SSMs in Algorithm 1, where the $|\cdot|$ and $\sqrt{\cdot}$ in Line 5 represent to element-wise absolute value and element-wise square root respectively. $[\cdot]_L$ extracts the last position of an element obtained from the convolution. The $\mathsf{Mean}(\cdot)$ operation in Line 6 calculates the mean value of a vector.

---

**Algorithm 1** Training one-layer SSMs with the initialization scheme (8)

---

**Input:** Training sequences $x_1, \ldots, x_n \in \mathbb{R}^{L \times d}$ with length $L$ and dimension $d$, a SSM initialization $\theta_0 = (C, A, B)$, a SSM kernel function $k(\theta) \in \mathbb{R}^{L \times d}$, number of epochs $s$
1: **for** $i = 0$ to $s - 1$ **do**
2:      **if** $i = 0$ **then**
3:          Sample a minibatch sequence $x = (x^{(1)}, \ldots, x^{(B)}) \in \mathbb{R}^{B \times L \times d}$
4:          Compute the mean $\mu \in \mathbb{R}^{L \times d}$ and variance $K \in \mathbb{R}^{L \times d}$ for $x$ along the batch dimension
5:          Compute $\tau(\theta_i)$ via convolution: $\tau(\theta_i) \leftarrow \left[ |k(\theta_i)| * \sqrt{K} + |k(\theta_i) * \mu| \right]_L \in \mathbb{R}^d$
6:          Average over the feature dimension: $\tau(\theta_i) \leftarrow \mathsf{Mean}^2(\tau(\theta_i))$
7:          Rescale by the initialization scheme (8): $\tilde{C} \leftarrow C/\sqrt{\tau(\theta_i)}$
8:          Start to train with the updated initialization $(\tilde{C}, A, B)$
9:      **end if**
10:      Regular training procedure
11: **end for**
**Output:** Updated model parameter $\theta_s$

---

## 4.3 GENERALIZATION BOUND AS A REGULARIZATION METHOD

In addition to its role as an initialization scheme, the generalization measure can also be regarded as a regularizer. In this section, we utilize the bound (3) to design a regularization method to improve the generalization performance, and simultaneously bring a little extra computational cost. For the generalization bound (3), we consider to use the dominant term (for large $T$) $\tau(\theta)$ defined in (7) as a regularizer. Then, the new empirical risk with regularization is given by

$$\tilde{R}_n(\theta) := R_n(\theta) + \lambda \cdot \tau(\theta), \tag{9}$$

where $\lambda \geq 0$ is the regularization coefficient. When training multi-layer SSMs, we calculate the complexity $\tau(\theta)$ in (9) at each layer and add them together as a total regularization. We describe the training procedures for one-layer SSMs in Algorithm 2, where the notations follow Algorithm 1.

---

**Algorithm 2** Training one-layer SSMs with the regularization method (9)

---

**Input:** Training sequences $x_1, \ldots, x_n \in \mathbb{R}^{L \times d}$ with length $L$ and dimension $d$, a SSM initialization $\theta_0$, a SSM kernel function $k(\theta) \in \mathbb{R}^{L \times d}$, loss function $\tilde{R}(\cdot, \cdot) : \mathbb{R}^d \times \mathbb{R}^d \to \mathbb{R}$, regularization coefficient $\lambda$, optimizer $\mathsf{OPT}$, number of epochs $s$
1: **for** $i = 0$ to $s - 1$ **do**
2:      Sample a minibatch input $x = (x^{(1)}, \ldots, x^{(B)}) \in \mathbb{R}^{B \times L \times d}$ with labels $(y^{(1)}, \ldots, y^{(B)})$
3:      Calculate the mean $\mu \in \mathbb{R}^{L \times d}$ and variance $K \in \mathbb{R}^{L \times d}$ for $x$ along the batch dimension
4:      Compute the SSM output via convolution: $y \leftarrow [k(\theta_i) * x]_L \in \mathbb{R}^{B \times d}$
5:      Compute the regularization via convolution: $\tau(\theta_i) \leftarrow \left[ |k(\theta_i)| * \sqrt{K} + |k(\theta_i) * \mu| \right]_L \in \mathbb{R}^d$
6:      Average over the feature dimension: $\tau(\theta_i) \leftarrow \mathsf{Mean}^2(\tau(\theta_i))$
7:      Compute the total loss $\mathcal{L} \leftarrow \frac{1}{B} \sum_{i=1}^{B} \tilde{R}(y_i, y^{(i)}) + \lambda \cdot \tau(\theta_i)$
8:      Parameters update: $\theta_{i+1} \leftarrow \mathsf{OPT}(\theta_i, \mathcal{L})$
9: **end for**
**Output:** Updated model parameter $\theta_s$

---

**Computational cost analysis.** From the training procedures in Algorithm 2, we can see that the newly introduced training complexity mainly comes from the calculation for the convolution between the SSM kernel and the sequence statistics $(\mu, K)$. Since the convolution can be conducted by the fast Fourier transform (Gu et al., 2022a) with complexity $\mathcal{O}(BdL \log L)$, then the new complexity for Algorithm 2 becomes $\mathcal{O}((B + 2)dL \log L)$, which is acceptable in the practical training.

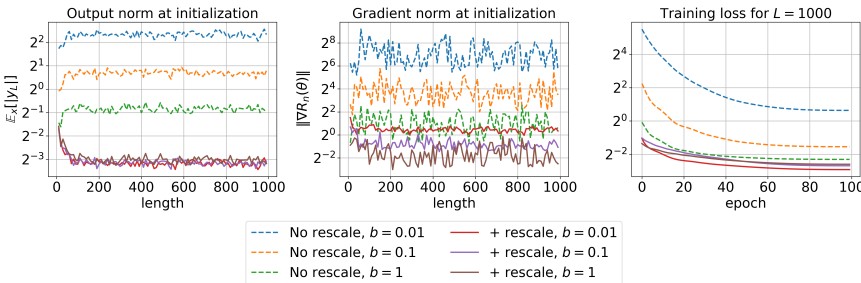

Figure 2: Effects of the initialization scheme (8) on the model output, the gradient norm and the optimization under different temporal dependencies. (Left) The output $\mathbb{E}_x[\|y_L\|]$ at initialization w.r.t. the Gaussian white noise sequence $(x_1, \ldots, x_L)$ for length $L$ from 1 to 1000; (Middle) The gradient norm $\|\nabla R_n(\theta)\|$ at initialization w.r.t. the mean squared error (MSE) for varied sequence length; (Right) The training MSE curve for the Gaussian white noise with length $L = 1000$.

| | Training loss (MSE) | | | | Test loss (MSE) | | | |
|---|---|---|---|---|---|---|---|---|
| | w/o (8, 9) | w (8) | w (9) | w (8, 9) | w/o (8, 9) | w (8) | w (9) | w (8, 9) |
| $b = 1$ | $0.21_{\pm 0.02}$ | $\mathbf{0.16}_{\pm 0.02}$ | $0.28_{\pm 0.02}$ | $0.24_{\pm 0.02}$ | $0.54_{\pm 0.04}$ | $0.57_{\pm 0.07}$ | $0.42_{\pm 0.009}$ | $\mathbf{0.42}_{\pm 0.02}$ |
| $b = 0.1$ | $0.35_{\pm 0.01}$ | $\mathbf{0.16}_{\pm 0.007}$ | $0.51_{\pm 0.01}$ | $0.32_{\pm 0.01}$ | $1.00_{\pm 0.05}$ | $1.20_{\pm 0.12}$ | $0.71_{\pm 0.03}$ | $\mathbf{0.65}_{\pm 0.02}$ |
| $b = 0.01$ | $1.20_{\pm 0.33}$ | $\mathbf{0.14}_{\pm 0.02}$ | $1.94_{\pm 0.33}$ | $0.32_{\pm 0.04}$ | $3.37_{\pm 0.50}$ | $1.43_{\pm 0.13}$ | $2.22_{\pm 0.16}$ | $\mathbf{0.67}_{\pm 0.03}$ |

Table 1: Training and test loss on the Gaussian white noise sequences with different coefficients $b$ after convergence. By adding the initialization scheme (8), SSMs achieve better optimization performance and are more robust on the final training loss value across different temporal dependencies. By adding the regularization term (9), SSMs get better generalization performance.

## 4.4 EXPERIMENTS

This section contains experiments to demonstrate the effectiveness of the proposed initialization scheme (8) and the regularization method (9). We use a synthetic sequence dataset and the Long Range Arena (LRA) benchmark (Tay et al., 2021) for numerical validations. To simplify the notation, we use w/o (8, 9), w (8), w (9) and w (8, 9) to represent the original base model, model trained with rescaling, model trained with regularization and model trained with both methods respectively.

**A synthetic dataset.** We consider a synthetic sequence dataset generated by a centered Gaussian white noise with the covariance function $K(s, t) = \frac{1}{|b|\sqrt{\pi}} e^{-((s-t)/b)^2}$, which is a stationary Gaussian process and satisfies Assumption 1 (ref Section C). Then we can get different temporal dependencies by varying the coefficient $b$, i.e., as the magnitude of $b$ decreasing, the temporal dependence of the corresponding Gaussian white noise decreases as well. In particular, as $b \to 0$, $\frac{1}{|b|\sqrt{\pi}} e^{-(x/b)^2}$ becomes a delta function $\delta(x)$, entailing a zero temporal dependence for the sequence data.

In the following experiment, we generate the sequence data by the Gaussian white noise with $b = [1, 0.1, 0.01]$. For each input sequence $(x_1, \ldots, x_L)$, its corresponding label is obtained by $\sin(x_{[L/2]})$, i.e., the sine value of the time-lagged input. We use the unidirectional S4-Legs model (Gu et al., 2022a) (that only contains the convolution layer) to train the sequence data. More details about the experiment setup are provided in Appendix A.1. In Figure 2, we plot the model output $\mathbb{E}_x[\|y_L\|]$ and the gradient norm $\|\nabla R_n(\theta)\|$ at initialization, and the training loss (w (8)) with different temporal patterns by varying the Gaussian white noise parameter $b$. We see that the initialization scheme (8) enhances the robustness of the output value scales(matches with Proposition 1), gradient norm at initialization and also the training loss value across different temporal structures. By comparing the final training loss with and without (8) in Table 1 (w/o (8, 9) vs w (8) and w (9) vs w (8, 9)), we see that adding the rescaling (8) also improves the training performance and makes the final training error more robust on different temporal dependencies (by varying $b$). For the regularization method (9), we compare the final test loss with and without (9) in Table 1 (w/o (8, 9) vs w (9) and w (8) vs w (8, 9)). We can see that the our regularization method improves the generalization performance. Moreover, combining (8) and (9), the model get the best test performance across various temporal structures of the sequence data.

|  |  | ListOps | Text | Retrieval | Image | Pathfinder | PathX |
|---|---|---|---|---|---|---|---|
| unidirectional S4-Legs | w/o (8, 9) | 59.45 | 79.27 | 88.28 | 87.99 | 87.84 |  |
|  | w (8) | 60.30 | 81.44 | 89.38 | 88.11 | 87.95 |  |
|  | w (9) | **60.65** | 81.45 | 89.21 | 87.79 | **90.36** |  |
|  | w (8, 9) | 60.40 | **82.56** | **90.13** | **88.28** | 90.03 |  |
|  | Time / epoch, w/o (8, 9) | 2min 57s | 29s | 21min | 1min 57s | 2min 52s |  |
|  | Time / epoch, w (9) | 3min 15s | 32s | 23min | 2min 11s | 3min 13s |  |
| bidirectional S4-Legs | w/o (8, 9) | **62.45** | 89.09 | 91.31 | 89.32 | 95.75 | $86.17_{\pm 1.74}$ |
|  | w (8) | 61.90 | 88.90 | 91.44 | 89.53 | 95.43 | $89.67_{\pm 0.18}$ |
|  | w (9) | 61.09 | **89.27** | 91.32 | **89.95** | 95.80 | $\mathbf{90.21}_{\pm 0.16}$ |
|  | w (8, 9) | 61.79 | 89.19 | **91.46** | 89.80 | **95.86** | $89.85_{\pm 0.72}$ |
|  | Time / epoch, w/o (8, 9) | 4min 40s | 3min 08s | 18min 20s | 2min 35s | 4min 06s | 13min 25s |
|  | Time / epoch, w (9) | 5min 18s | 3min 34s | 20min 30s | 2min 46s | 4min 26s | 14min 40s |
| bidirectional S4D-Legs | w/o (8, 9) | 57.80 | 83.91 | 90.84 | 86.47 | 87.36 | $90.19_{\pm 0.78}$ |
|  | w (8) | 57.25 | 84.79 | 91.01 | 86.34 | 88.35 | $\mathbf{90.25}_{\pm 0.15}$ |
|  | w (9) | 57.50 | 84.52 | **91.08** | 87.33 | 87.27 | $90.19_{\pm 0.34}$ |
|  | w (8, 9) | **58.45** | **85.75** | 91.04 | 87.28 | **88.96** | $89.46_{\pm 1.21}$ |
|  | Time / epoch, w/o (8, 9) | 1min 47s | 51s | 18min 15s | 1min 27s | 1min 26s | 10min |
|  | Time / epoch, w (9) | 2min | 55s | 22min 36s | 1min 50s | 1min 50s | 11min 11s |

Table 2: Test accuracy and running time (per epoch in A100 GPU) on the LRA benchmark under different settings for different models. The unidirectional model processes a sequence in one direction while the bidirectional model consists of two separate layers that process the sequence in opposite directions. Mean and standard error for the PathX results are reported based on 3 independent runs.

**LRA benchmark.** We investigate the effects of the initialization scheme (8) and the regularization method (9) on the LRA benchmark, We consider three base models: unidirectional S4-Legs (Gu et al., 2022a), bidirectional S4-Legs (Goel et al., 2022) and bidirectional S4D-Legs (Gu et al., 2022b). Among these three models, the unidirectional S4-Legs is the one that is closest to our model setting (1), however, it performs poorly in challenge datasets. Thus, we do not use the unidirectional S4-Legs to train the PathX. We follow the training rules as described by Gu et al. (2023), but with adjustments to the model size. For example, the model sizes used to train the PathX for both S4-Legs and S4D-Legs are relatively small compared with the ones used in Gu et al. (2023) to save training time. More details on the dataset description and the experiment setup are given in Appendix A.2.

By comparing the test accuracy for w/o (8, 9) vs w (9) and w (8) vs w (8, 9) in Table 2, we see that adding the regularization (9) enhances the generalization performance in most cases for all three models. In particular, when combining the initialization scheme (8) and the regularization (9), one get the best test performance in half of tasks, indicating that our proposed optimization designs effectively improve the generalization performance. We also compare the running time without or with the proposed optimization designs. Since (8) is conducted before training which will not introduce additional training complexity, we report the running time for w/o (8, (9)) and w (9) in Table 2. The results show that the regularization brings a little extra computational cost, matching the computational cost analysis in Section 4.3. We include an ablation study for the hyperparameter $\lambda$ and add more experiment results in Appendix A.2.

## 5 DISCUSSION

In this work, we study the optimization and the generalization for SSMs. Specifically, we give a data-dependent generalization bound, revealing an effect of the temporal dependencies of the sequence data on the generalization. Based on the bound, we design two algorithms: a new initialization scheme and a regularization method, to improve the optimization and generalization for SSMs. There are still some gaps between the theory and the methodologies in this paper. The first one that the skip connection matrix $D$ is omitted in our defined model (1). This will not affect our generalization bound because we may express the explicit solution for (1) as $y(t) = \int_0^t (\rho_\theta(s) + D\delta(s))x(t-s)ds$ where $\delta(\cdot)$ is a delta function, which is still a convolution model with a new kernel $\rho_\theta(s) + D\delta(s)$. However, the initialization scheme (8) only adjusts $C$ and requires the kernel function to be linear in $C$. Hence, (8) may not work well when $Dx(t)$ is much larger than $\int_0^t \rho_\theta(s)x(t-s)ds$. The second gap is that our theory is for single-layer linear SSMs. When nonlinearities are added, our generalization bound still works for single-layer SSMs if the nonlinearity does not affect the Hölder condition and the sub-Gaussian property (Assumption 1). For Lipschitz (also Hölder continuous) functions, there are some known examples (see Appendix G) where the sub-Gaussian condition remains after the nonlinearity. The extension of our theory to the multi-layer case is an interesting direction, which we leave for future work.

**Reproducibility** The the generalization bound (2) for linear regression is proved in Appendix B. The proof for Theorem 1 is provided in Appendix E. The derivations for (5) and (6) in Section 4.1 are given in Appendix D. The proof for Proposition 1 is in Appendix F. The details for the experiment settings are shown in Appendix A.1 and Appendix A.2.

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

| | $D$ | $H$ | $N$ | Dropout | Learning rate | Batch size | Epochs | Weight decay |
|---|---|---|---|---|---|---|---|---|
| ListOps | 6 | 128 | 64 | 0 | 0.01 | 50 | 50 | 0.01 |
| Text | 4 | 128 | 64 | 0 | 0.01 | 50 | 40 | 0 |
| Retrieval | 6 | 256 | 64 | 0 | 0.002 | 64 | 25 | 0 |
| Image | 6 | 512 | 64 | 0.2 | 0.004 | 50 | 100 | 0.01 |
| Pathfinder | 6 | 256 | 64 | 0.1 | 0.004 | 100 | 200 | 0 |

Table 3: List of the unidirectional S4-Legs model hyperparameters for the LRA benchmark.

## A  EXPERIMENTS DETAILS

In this section, we provide more details for the experiments of the synthetic dataset and the LRA benchmark in Section 4.4.

### A.1  THE SYNTHETIC EXPERIMENT

For the Gaussian white noise sequences, we generate 100 i.i.d. sequences for training and 1000 i.i.d. sequences for test. The timescale for the discrete sequences is set to be 1, i.e., to generate a Gaussian white noise sequence with length $L$, we sample from a multivariate normal distribution with mean zero and covariance matrix $K_{i,j} = h(i - j)$ for $i, j \in [1 : L]$, where $h(t) = \frac{1}{|b|\sqrt{\pi}}e^{-(t/b)^2}$. The model that we use is the one-layer S4 model that only contains the FFTConv (fast Fourier transform convolution) layer and without activation and the skip connection ($D = 0$) (Gu et al., 2022a). The state space dimension for the FFTConv layer is 64, other settings such as the discretization, the initialization and the parameterization follow the default settings in Gu et al. (2023), i.e., we use the ZOH discretization, the LegS initialization and the exponential parameterization for the hidden state matrix $A$.

For the optimizer, we follow Gu et al. (2023) to set the optimizer by groups. For the (ZOH) timescale $\Delta$, the hidden state matrices $A, B$, we use Adam optimizer with learning rate 0.001, while for the matrix $C$, we use AdamW with learning rate 0.01 and decay rate 0.01. For all the parameters, we use the cosine annealing schedule. The batch size is set to be 100 (full batch) and the training epochs is 100. The regularization coefficient $\lambda$ used for training with (9) is set to be 0.01 across all the temporal patterns.

### A.2  LRA BENCHMARK

**Datasets.** The datasets in the LRA benchmark contain (1) ListOps (Nangia & Bowman, 2018), a dataset that is made up of a list of mathematical operations with answers; (2) Text (Maas et al., 2011), a movie review dataset collected from IMDB, which is used for sentiment analysis; (3) Retrieval (Radev et al., 2009), a task of retrieving documents utilizing byte-level texts from the ACL Anthology Network. (4)Image (Krizhevsky et al., 2009), a sequential CIFAR10 dataset used for sequence classification; (5) Pathfinder (Linsley et al., 2018), a task that requires a model to tell whether two points in an image are connected by a dashed path. (6) PathX, a similar but more challenge task as Pathfinder with a higher image resolution increased from $32 \times 32$ to $128 \times 128$.

**Models.** The models consist of unidirectional S4-Legs, bidirectional S4-Legs and bidirectional S4D-Legs. The unidirectional model processes the input sequence in one direction from past to future, while the bidirectional model processes the input sequence in two directions from past to future and from future to past. All the three models use the default Legs initialization. Discretization and model parameterization are set to be consistent with Gu et al. (2023). For the optimizer, we also follow the standard setup in Gu et al. (2023) that the hidden state matrices are trained in a relatively small learning rate with no weight decay, while other parameters are trained with AdamW with a larger learning rate. Let $D, H, N$ denote the depth, feature dimension and hidden state space dimension respectively, we summarize the model hyperparameters for unidirectional S4-Legs, bidirectional S4-Legs and bidirectional S4D-Legs in Table 3, 4 and 5 respectively.

**Ablation studies on $\lambda$.** When training with the regularization method (9), we vary the regularization coefficient $\lambda$ for different magnitudes ranging from $10^{-6}$ to $10^{-1}$ when the model performs

| | $D$ | $H$ | $N$ | Dropout | Learning rate | Batch size | Epochs | Weight decay |
|---|---|---|---|---|---|---|---|---|
| ListOps | 6 | 256 | 4 | 0 | 0.01 | 32 | 40 | 0.05 |
| Text | 6 | 256 | 4 | 0 | 0.01 | 16 | 32 | 0.05 |
| Retrieval | 6 | 256 | 4 | 0 | 0.01 | 64 | 20 | 0.05 |
| Image | 6 | 512 | 64 | 0.1 | 0.01 | 50 | 200 | 0.05 |
| Pathfinder | 6 | 256 | 64 | 0.0 | 0.004 | 64 | 200 | 0.05 |
| PathX | 4 | 96 | 64 | 0.0 | 0.0005 | 64 | 50 | 0.05 |

Table 4: List of the bidirectional S4-Legs model hyperparameters for the LRA benchmark.

| | $D$ | $H$ | $N$ | Dropout | Learning rate | Batch size | Epochs | Weight decay |
|---|---|---|---|---|---|---|---|---|
| ListOps | 4 | 128 | 64 | 0 | 0.01 | 50 | 40 | 0.05 |
| Text | 4 | 128 | 64 | 0 | 0.01 | 50 | 50 | 0 |
| Retrieval | 6 | 256 | 64 | 0 | 0.01 | 64 | 20 | 0.05 |
| Image | 6 | 512 | 64 | 0.1 | 0.01 | 50 | 200 | 0.05 |
| Pathfinder | 6 | 128 | 64 | 0.0 | 0.004 | 64 | 40 | 0.01 |
| PathX | 4 | 96 | 64 | 0.0 | 0.0005 | 64 | 50 | 0.05 |

Table 5: List of the bidirectional S4D-Legs model hyperparameters for the LRA benchmark.

best on the validation set. In Table 6, 7 and 8, we report the test accuracy on the LRA benchmark with different $\lambda$ for unidirectional S4-Legs, bidirectional S4-Legs and bidirectional S4D-Legs respectively. From the results in Table 6, 7 and 8, we find that for all the three models, the test performance is much more sensitive to the regularization coefficient $\lambda$ for the Pathfinder and PathX tasks compared to other tasks. To investigate the magnitude of the optimal $\lambda$, we plot the generalization measure (7) during the training process for the three models in Figure 3, 4, 5 respectively. As shown in these figures, the generalization measure when training the Pathfinder and the PathX datasets without regularization is much larger than other datasets for each single model, making the generalization measure very sensitive to the magnitude of $\lambda$, thus it should be set with a relatively small value. When comparing the generalization measure across these three models for each individual dataset, we find that the bidirectional S4-Legs has smaller generalization measure compared with unidirectional S4-Legs and bidirectional S4D-Legs. This validates that our generalization measure captures the test performance in the sense that bidirectional S4-Legs performs better than the other two models.

## B  PROOF FOR THE LINEAR REGRESSION RESULT IN SECTION 3.2.

In this section, we give the proof for the generalization bound (2). The proof is based on the following uniform-convergence generalization bound in Mohri et al. (2012).

**Lemma 1.** *Consider a family of functions $\mathcal{F}$ mapping from $\mathcal{Z}$ to $[a, b]$. Let $\mathcal{D}$ denote the distribution according to which samples are drawn. Then for any $\delta > 0$, with probability at least $1 - \delta$ over the draw of an i.i.d. sample $S = \{z_1, \ldots, z_n\}$, the following holds for all $f \in \mathcal{F}$:*

$$\mathbb{E}_{z \sim \mathcal{D}}\left[f(z)\right] - \frac{1}{n}\sum_{i=1}^{n} f(z_i) \leq 2\mathcal{R}_S(\mathcal{F}) + 3(b - a)\sqrt{\frac{\log(2/\delta)}{2n}},$$

| | ListOps |
|---|---|
| $\lambda = 0$ | 59.45 |
| $\lambda = 10^{-3}$ | **60.65** |
| $\lambda = 10^{-2}$ | 59.35 |
| $\lambda = 10^{-1}$ | 57.60 |

| | Text |
|---|---|
| $\lambda = 0$ | 79.27 |
| $\lambda = 10^{-3}$ | 80.74 |
| $\lambda = 10^{-2}$ | **81.45** |
| $\lambda = 10^{-1}$ | 80.45 |

| | Retrieval |
|---|---|
| $\lambda = 0$ | 88.28 |
| $\lambda = 10^{-3}$ | **89.21** |

| | Image |
|---|---|
| $\lambda = 0$ | **87.99** |
| $\lambda = 10^{-5}$ | 87.79 |
| $\lambda = 10^{-3}$ | 87.70 |
| $\lambda = 10^{-2}$ | 87.62 |

| | Pathfinder |
|---|---|
| $\lambda = 0$ | 87.84 |
| $\lambda = 10^{-5}$ | **90.36** |
| $\lambda = 10^{-4}$ | 50 |

Table 6: Test accuracy for unidirectional S4-Legs on LRA benchmark by varying the regularization coefficient $\lambda$.

|  | ListOps |
|---|---|
| $\lambda = 0$ | **62.45** |
| $\lambda = 10^{-4}$ | 60.69 |
| $\lambda = 10^{-3}$ | 61.09 |

|  | Text |
|---|---|
| $\lambda = 0$ | 89.09 |
| $\lambda = 10^{-4}$ | **89.27** |
| $\lambda = 10^{-3}$ | 89.17 |

|  | Retrieval |
|---|---|
| $\lambda = 0$ | 91.31 |
| $\lambda = 10^{-4}$ | 91.14 |
| $\lambda = 5 \times 10^{-4}$ | **91.32** |
| $\lambda = 10^{-3}$ | 91.23 |
| $\lambda = 5 \times 10^{-3}$ | 91.07 |

|  | Image |
|---|---|
| $\lambda = 0$ | 89.32 |
| $\lambda = 10^{-4}$ | 89.47 |
| $\lambda = 10^{-3}$ | **89.95** |
| $\lambda = 5 \times 10^{-3}$ | 89.85 |
| $\lambda = 10^{-2}$ | 89.62 |

|  | Pathfinder |
|---|---|
| $\lambda = 0$ | 95.75 |
| $\lambda = 10^{-5}$ | **95.80** |
| $\lambda = 10^{-4}$ | 95.79 |
| $\lambda = 10^{-3}$ | 50 |

|  | PathX |
|---|---|
| $\lambda = 0$ | 86.82 |
| $\lambda = 10^{-5}$ | 89.17 |
| $\lambda = 10^{-4}$ | **90.09** |
| $\lambda = 10^{-3}$ | 50 |

Table 7: Test accuracy for bidirectional S4-Legs on LRA benchmark by varying the regularization coefficient $\lambda$.

|  | ListOps |
|---|---|
| $\lambda = 0$ | **57.80** |
| $\lambda = 10^{-5}$ | 56.60 |
| $\lambda = 10^{-4}$ | 57.20 |
| $\lambda = 10^{-3}$ | 57.50 |
| $\lambda = 10^{-2}$ | 57.25 |

|  | Text |
|---|---|
| $\lambda = 0$ | 83.91 |
| $\lambda = 10^{-4}$ | 84.27 |
| $\lambda = 10^{-3}$ | 84.35 |
| $\lambda = 10^{-2}$ | **84.52** |

|  | Retrieval |
|---|---|
| $\lambda = 0$ | 90.84 |
| $\lambda = 10^{-5}$ | **91.08** |
| $\lambda = 10^{-4}$ | 90.94 |

|  | Image |
|---|---|
| $\lambda = 0$ | 86.47 |
| $\lambda = 10^{-4}$ | **87.33** |

|  | Pathfinder |
|---|---|
| $\lambda = 0$ | **87.36** |
| $\lambda = 10^{-6}$ | 87.27 |
| $\lambda = 10^{-5}$ | 87.25 |
| $\lambda = 5 \times 10^{-5}$ | 50 |

|  | PathX |
|---|---|
| $\lambda = 0$ | **90.35** |
| $\lambda = 10^{-6}$ | 88.72 |
| $\lambda = 10^{-5}$ | 90.19 |
| $\lambda = 10^{-4}$ | 50 |

Table 8: Test accuracy for bidirectional S4D-Legs on LRA benchmark by varying the regularization coefficient $\lambda$.

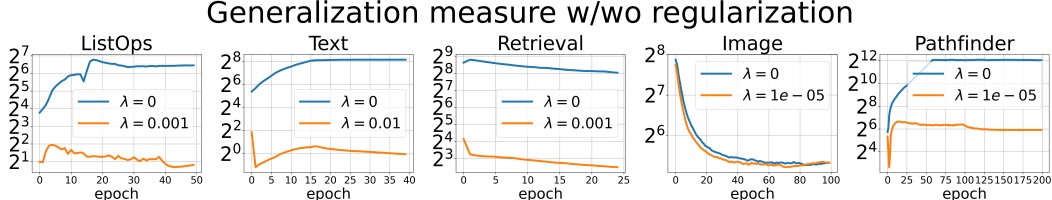

Figure 3: Generalization measure (7) of unidirectional S4-Legs on LRA benchmark with or without regularization for different regularization coefficient $\lambda$ in (9).

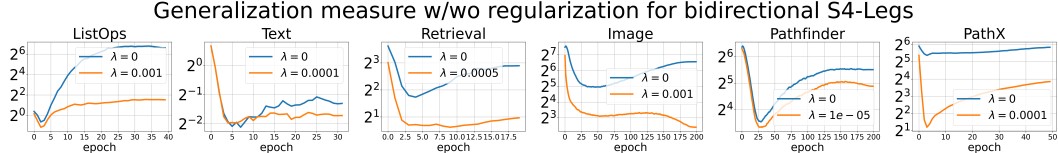

Figure 4: Generalization measure (7) of bidirectional S4-Legs on LRA benchmark with or without regularization for different regularization coefficient $\lambda$ in (9).

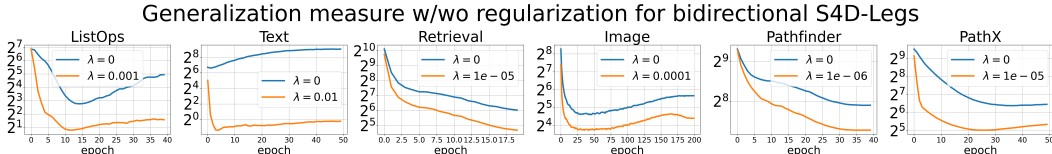

Figure 5: Generalization measure (7) of bidirectional S4D-Legs on LRA benchmark with or without regularization for different regularization coefficient $\lambda$ in (9).

where $\mathcal{R}_S(\mathcal{F})$ is the empirical Rademacher complexity with respect to the sample $S$, defined as: $\mathcal{R}_S(\mathcal{F}) = \mathbb{E}_\sigma \left[ \sup_{f \in \mathcal{F}} \frac{1}{n} \sum_{i=1}^n \sigma_i f(z_i) \right]$. $\{\sigma_i\}_{i=1}^n$ are i.i.d. random variables drawn from $U\{-1, 1\}$ with $P(\sigma_i = 1) = P(\sigma_i = -1) = 0.5$.

And the Talagrand's contraction lemma Ledoux & Talagrand (2013).

**Lemma 2.** *Let $H$ be a hypothesis set of functions mapping $\mathcal{X}$ to $\mathbb{R}$ and $\Psi_1, \ldots, \Psi_m$, $\mu$-Lipschitz functions for some $\mu > 0$. Then, for any sample $S$ of $m$ points $x_1, ..., x_m \in \mathcal{X}$, the following inequality holds*

$$\frac{1}{m} \mathbb{E}_\sigma \left[ \sup_{h \in H} \sum_{i=1}^m \sigma_i \left( \Psi_i \circ h \right)(x_i) \right] \leq \frac{\mu}{m} \mathbb{E}_\sigma \left[ \sup_{h \in H} \sum_{i=1}^m \sigma_i h(x_i) \right]$$

Now we begin our proof:

*Proof.* First, notice for any $i \in [1 : n]$ and $\theta \in \Theta$, we have

$$(\theta^\top x_i - y_i)^2 \leq 2(\theta^\top x_i)^2 + 2y_i^2 \leq 2r^2 R^2 + 2$$

Second, note that $(\theta^\top x_i - y_i)^2$ is $2 \sup_{\theta \in \Theta, i \in [1:n]} |\theta^\top x_i - y_i|$-Lipschitz (the maximum gradient norm) with respect to $\theta^\top x_i - y_i$, and we can bound the Lipschitz constant as

$$2 \sup_{\theta \in \Theta, i \in [1:n]} |\theta^\top x_i - y_i| \leq 2rR + 2$$

Then by Lemma 2, the Rademacher complexity for the linear model is bounded as

$$\begin{aligned}
\mathcal{R}_S(\mathcal{F}) &= \frac{1}{n} \mathbb{E}_\sigma \left[ \sup_{\|\theta\|_2 \leq R} \sum_{i=1}^n \sigma_i (\theta^\top x_i - y_i)^2 \right] \\
&\leq \frac{2rR+2}{n} \mathbb{E}_\sigma \left[ \sup_{\|\theta\|_2 \leq R} \sum_{i=1}^n \sigma_i (\theta^\top x_i - y_i) \right] \\
&= \frac{2rR+2}{n} \mathbb{E}_\sigma \left[ \sup_{\|\theta\|_2 \leq R} \sum_{i=1}^n \sigma_i \theta^\top x_i \right] \\
&\leq \frac{2R(rR+1)}{n} \mathbb{E}_\sigma \left\| \sum_{i=1}^n \sigma_i x_i \right\| \\
&\leq \frac{2R(rR+1)}{n} \sqrt{\mathbb{E}_\sigma \left\| \sum_{i=1}^n \sigma_i x_i \right\|^2} \\
&= \frac{2R(rR+1)}{n} \sqrt{\sum_{i=1}^n \|x_i\|^2} \\
&\leq \frac{2rR(rR+1)}{\sqrt{n}}
\end{aligned}$$

Combining with the function value bound, we get the desired bound (2) by Lemma 1. $\qquad\square$

## C  DETAILED DISCUSSIONS OF ASSUMPTION 1

In this section, we add more discussions on the Assumption 1 and provide some concrete examples for the stochastic processes that satisfy the assumption. We first write down the complete description for the Kolmogorov continuity theorem.

**Lemma 3** (Kolmogorov). *Let $\{X_t\}_{t\geq 0}$ be a real-valued stochastic process such that there exists positive constants $\alpha, \beta, C$ satisfying*

$$\mathbb{E}\left[|X_t - X_s|^\alpha\right] \leq C|t - s|^{1+\beta}$$

*for all $s, t \geq 0$. Then $X$ has a continuous modification which, with probability one, is locally $\gamma$-Hölder continuous for every $0 < \gamma < \beta/\alpha$.*

In the case of Brownian motion on $\mathbb{R}$, the choice of constants $\alpha = 4, \beta = 1, C = 2$ will work in the Kolmogorov continuity theorem. When it comes to the Gaussian process, we have the following theorem (Azmoodeh et al., 2014, Theorem 1.) that gives a necessary and sufficient condition for Hölder continuity.

**Lemma 4.** *A centered (mean zero) Gaussian process $X$ is Hölder continuous of any order $a < H$, i.e.,*

$$|X_t - X_s| \leq C_\varepsilon |t - s|^{H-\varepsilon}, \quad \forall \varepsilon \in (0, H)$$

*if and only if there exists constants $c_\varepsilon$ such that*

$$\mathbb{E}\left[(X_t - X_s)^2\right] \leq c_\varepsilon (t - s)^{2H-2\varepsilon}, \quad \forall \varepsilon \in (0, H)$$

For a stationary Gaussian process with covariance $K(s - t)$, the Hölder continuity (in expectation) assumption is equivalent to $1 - K(s - t)/K(0) \leq c_\alpha(t - s)^{2\alpha}/2$ for any $\alpha \in (0, H)$. Now combining these results, we see that for any stationary Gaussian process with continuous mean $\mu(t)$, covariance $K(s - t)$, and $1 - K(s - t)/K(0) \leq c_\alpha(t - s)^{2\alpha}/2, \forall \alpha \in (0, H)$, it satisfies Hölder continuity in Assumption 1. As for the sub-Gaussian property, since the normalized Gaussian process $\tilde{X}_t$ is standard normal at each time $t$, then any Gaussian process that satisfies Hölder continuity automatically satisfies the sub-Gaussian property in Assumption 1. Concrete examples include:

- identical sequences: $x(t) = x$ for all $t \in [0, T]$, where $x \sim \mathcal{N}(0, 1)$

- Gaussian white noise: $\mu(t) = 0, K(s, t) = \frac{1}{|b|\sqrt{\pi}} e^{-((s-t)/b)^2}$ for some $b \neq 0$

- Ornstein-Uhlenbeck process: $\mu(t) = 0, K(s, t) = e^{-|s-t|}$

**Relaxations of Assumption 1.** In fact, Assumption 1 is used to show upper bounds for two key terms (14) in the proof of Theorem 1. In particular, the sub-Gaussian property in Assumption 1 guarantees that the input random process is bounded in a finite time set with high probability. The Hölder condition then ensures the boundedness in a infinite time set $t \in [0, T]$. Thus, if the input random process is from a finite subset of $\mathbb{R}$, then the Hölder condition can be removed. For example, in computer vision tasks when the input image is flattened as a sequence, the range for each pixel value is a finite set (for a MNIST image, each pixel value is a positive integer between 0 to 255). In that case, the Holder continuity condition in Assumption 1 can be dropped.

## D   DERIVATIONS FOR (5) AND (6) IN SECTION 4.1

For the left zero padding transformation, the key term in (3) becomes

$$\int_0^{2T} |\rho_\theta(2T - t)| \sqrt{K_1(t, t)} dt + \left|\int_0^{2T} \rho_\theta(2T - t)\mu_1(t) dt\right| + 1$$

$$= \int_0^T |\rho_\theta(T - t)| \sqrt{K(t, t)} dt + \left|\int_0^T \rho_\theta(T - t)\mu(t) dt\right| + 1$$

For the right zero padding transformation, the key term in (3) becomes

$$\int_0^{2T} |\rho_\theta(2T - t)| \sqrt{K_2(t, t)} dt + \left|\int_0^{2T} \rho_\theta(2T - t)\mu_2(t) dt\right| + 1$$

$$= \int_0^T |\rho_\theta(2T - t)| \sqrt{K(t, t)} dt + \left|\int_0^T \rho_\theta(2T - t)\mu(t) dt\right| + 1$$

$$= \int_0^T \left|Ce^{AT}e^{A(T-t)}B\right| \sqrt{K(t, t)} dt + \left|\int_0^T Ce^{AT}e^{A(T-t)}B\mu(t) dt\right| + 1$$

Then we get (5) and (6).

## E  PROOF FOR THEOREM 1

In this section, we will prove Theorem 1. Before moving into the formal proof, we first introduce some useful lemmas that help to build the proof.

The first lemma is the Massart Lemma for the Rademacher complexity with finite class.

**Lemma 5** (Massart). *Let $\mathcal{A}$ be some finite subset of $R^m$ and $\sigma_1, \ldots, \sigma_m$ be independent Rademacher random variables. Let $r = \sup_{a \in \mathcal{A}} \|a\|$. Then, we have,*

$$\mathbb{E}_\sigma \left[ \sup_{a \in \mathcal{A}} \sum_{i=1}^m \sigma_i a_i \right] \leq r\sqrt{2 \log |\mathcal{A}|}$$

The second lemma is to bound the supremum of a stochastic process that is Hölder continuous and sub-Gaussian.

**Lemma 6** (Hölder maximal inequality). *Suppose $\{X_t\}_{t \in [0,T]}$ is a centered Hölder process, i.e., $\exists L, H > 0, s.t. |X_s - X_t| \leq L|s-t|^H, \forall s, t \in [0, T]$. If further $X_t$ is $\sigma^2$-sub-Gaussian for every $t \in [0, T]$, i.e., $\forall u > 0, P(|X_t| \geq u) \leq 2\exp(-u^2/2\sigma^2)$ for some $\sigma > 0$. Then with probability at least $1 - \delta$,*

$$\sup_{t \in [0,T]} |X_t| \leq L + \sigma\sqrt{2 \log(2T/\delta)}.$$

*Proof.* The proof is based on the $\varepsilon$-net and covering number argument. We first discretize the time interval $[0, T]$ into $N$ parts $[0, T/N] \cup [T/N, 2T/N] \cdots \cup [(N-1)T/N, T]$. Then for any time $t \in [0, T]$, there exists a $\pi(t) \in \{0, T/N, \ldots, (N-1)T/N\}$ such that $|t - \pi(t)| \leq T/N$. Therefore, by Hölder continuity, we have

$$\sup_{t \in [0,T]} |X_t| \leq \sup_{t \in [0,T]} |X_t - X_{\pi(t)}| + \sup_{t \in [0,T]} |X_{\pi(t)}| \leq L\left(\frac{T}{N}\right)^H + \max_{i \in [0:N-1]} |X_{iT/N}|.$$

Since $X_t$ is sub-Guassian for every time $t \in [0, T]$, then for each $i \in [0 : N - 1]$, by letting $u = \sigma\sqrt{2 \log(2N/\delta)}$, we have with probability at least $1 - \delta/N$,

$$X_{iT/N} \leq \sigma\sqrt{2 \log(2N/\delta)}.$$

Taking the union bound over all $i \in [0 : N - 1]$, we have with probability at least $1 - \delta$,

$$\max_{i \in [0:N-1]} X_{iT/N} \leq \sigma\sqrt{2 \log(N/\delta)}.$$

Hence,

$$\sup_{t \in [0,T]} X_t \leq L\left(\frac{T}{N}\right)^H + \sigma\sqrt{2 \log(2N/\delta)}$$

holds for all $N$. Here we simply take $N = [T] + 1$, then we get

$$\sup_{t \in [0,T]} X_t \leq L + \sigma\sqrt{2 \log(2T/\delta)}.$$

$\square$

Now we are ready to prove the main result Theorem 1.

*Proof.* We let $g_\theta(x) := \int_0^T \rho_\theta(T - t)x(t)dt - y$, then the generalization gap is given by

$$R_x(\theta) - R_n(\theta) = \mathbb{E}_x[g_\theta^2(x)] - \frac{g_\theta^2(x_1) + \ldots + g_\theta^2(x_n)}{n}.$$

Now let hypothesis space $\mathcal{F} = \{x \mapsto g_\theta^2(x) : \theta \in \Theta\}$, then its empirical Rademacher complexity is given by

$$
\mathcal{R}_S(\mathcal{F}) = \mathbb{E}_\sigma \left[ \sup_{\theta \in \Theta} \frac{1}{n} \sum_{i=1}^n \sigma_i g_\theta^2(x_i) \right]
$$

$$
= \frac{1}{n} \mathbb{E}_\sigma \left[ \sup_{\theta \in \Theta} \sum_{i=1}^n \sigma_i \left| \int_0^T \rho_\theta(T-t) x_i(t) dt - y_i \right|^2 \right]
$$

By the Talagrand's contraction Lemma 2, since $g_\theta^2(x_i)$ is $2 \sup_{\theta \in \Theta, i \in [1:n]} |g_\theta(x_i)|$ Lipschitz, we have

$$
\mathcal{R}_S(\mathcal{F}) \leq 2 \sup_{\theta \in \Theta, i \in [1:n]} |g_\theta(x_i)| \cdot \frac{1}{n} \mathbb{E}_\sigma \left[ \sup_{\theta \in \Theta} \sum_{i=1}^n \sigma_i \left( \int_0^T \rho_\theta(T-t) x_i(t) dt - y_i \right) \right]
$$

$$
= \frac{2 \sup_{\theta \in \Theta, i \in [1:n]} |g_\theta(x_i)|}{n} \mathbb{E}_\sigma \left[ \sup_{\theta \in \Theta} \int_0^T \rho_\theta(T-t) \sum_{i=1}^n \sigma_i x_i(t) dt \right]
$$

Now we separate the expectation into two parts: the unbiased part invovled with $x_i(t) - \mu(t)$ and the biased part $\mu(t)$, by noticing that

$$
\mathbb{E}_\sigma \left[ \sup_{\theta \in \Theta} \int_0^T \rho_\theta(T-t) \sum_{i=1}^n \sigma_i x_i(t) dt \right]
$$

$$
= \mathbb{E}_\sigma \left[ \sup_{\theta \in \Theta} \int_0^T \rho_\theta(T-t) \sum_{i=1}^n \sigma_i (x_i(t) - \mu(t)) dt + \int_0^T \rho_\theta(T-t) \sum_{i=1}^n \sigma_i \mu(t) dt \right]
$$

$$
\leq \mathbb{E}_\sigma \left[ \sup_{\theta \in \Theta} \int_0^T \rho_\theta(T-t) \sum_{i=1}^n \sigma_i (x_i(t) - \mu(t)) dt \right] + \mathbb{E}_\sigma \left[ \sup_{\theta \in \Theta} \int_0^T \rho_\theta(T-t) \sum_{i=1}^n \sigma_i \mu(t) dt \right]
$$

For the unbiased part, by the Hölder's inequality, for any $p, q \in [1, \infty]$ such that $\frac{1}{p} + \frac{1}{q} = 1$,

$$
\mathbb{E}_\sigma \left[ \sup_{\theta \in \Theta} \int_0^T \rho_\theta(T-t) \sum_{i=1}^n \sigma_i (x_i(t) - \mu(t)) dt \right]
$$

$$
\leq \sup_{\theta \in \Theta} \left( \int_0^T |\rho_\theta^p(T-t)| \, K^{p/2}(t,t) dt \right)^{1/p} \mathbb{E}_\sigma \left[ \left( \int_0^T \left| \sum_{i=1}^n \sigma_i \frac{x_i(t) - \mu(t)}{\sqrt{K(t,t)}} \right|^q dt \right)^{1/q} \right] \tag{10}
$$

For the biased part,

$$
\mathbb{E}_\sigma \left[ \sup_{\theta \in \Theta} \int_0^T \rho_\theta(T-t) \sum_{i=1}^n \sigma_i \mu(t) dt \right] \leq \sup_{\theta \in \Theta} \left| \int_0^T \rho_\theta(T-t) \mu(t) dt \right| \mathbb{E}_\sigma \left[ \left| \sum_{i=1}^n \sigma_i \right| \right]
$$

$$
\leq \sup_{\theta \in \Theta} \left| \int_0^T \rho_\theta(T-t) \mu(t) dt \right| \sqrt{\mathbb{E}_\sigma \left[ \left| \sum_{i=1}^n \sigma_i \right|^2 \right]} \tag{11}
$$

$$
= \sqrt{n} \sup_{\theta \in \Theta} \left| \int_0^T \rho_\theta(T-t) \mu(t) dt \right|
$$

Now for the unbiased part (10), we take $p = 1, q = \infty$. Then we have

$$
\mathbb{E}_\sigma \left[ \sup_{\theta \in \Theta} \int_0^T \rho_\theta(T-t) \sum_{i=1}^n \sigma_i (x_i(t) - \mu(t)) dt \right]
$$

$$
\leq \sup_{\theta \in \Theta} \left( \int_0^T |\rho_\theta(T-t)| \sqrt{K(t,t)} dt \right) \mathbb{E}_\sigma \left[ \sup_{t \in [0,T]} \left| \sum_{i=1}^n \sigma_i \frac{x_i(t) - \mu(t)}{\sqrt{K(t,t)}} \right| \right] \tag{12}
$$

Also by the same argument, note that

$$
\sup_{\theta \in \Theta, i \in [1:n]} |g_\theta(x_i)|
$$

$$
= \sup_{\theta \in \Theta, i \in [1:n]} \left| \int_0^T \rho_\theta(T - t) x_i(t) dt - y_i \right|
$$

$$
\leq \sup_{\theta \in \Theta, i \in [1:n]} \left| \int_0^T \rho_\theta(T - t)(x_i(t) - \mu(t)) dt \right| + \sup_{\theta \in \Theta} \left| \int_0^T \rho_\theta(T - t) \mu(t) dt \right| + 1
$$

$$
\leq \sup_{\theta \in \Theta} \left( \int_0^T |\rho_\theta(T - t)| \sqrt{K(t,t)} dt \right) \sup_{i \in [1:n], t \in [0,T]} \left| \frac{x_i(t) - \mu(t)}{\sqrt{K(t,t)}} \right| + \sup_{\theta \in \Theta} \left| \int_0^T \Re(\rho_\theta(T - t)) \mu(t) dt \right| + 1
$$

$$(13)$$

Thus, there are two terms that we need to bound:

$$
\sup_{i \in [1:n], t \in [0,T]} \left| \frac{x_i(t) - \mu(t)}{\sqrt{K(t,t)}} \right|, \quad \mathbb{E}_\sigma \left[ \sup_{t \in [0,T]} \left| \sum_{i=1}^n \sigma_i \frac{x_i(t) - \mu(t)}{\sqrt{K(t,t)}} \right| \right] \tag{14}
$$

For the first term, notice that the normalized Gaussian process $\frac{x_i(t) - \mu(t)}{\sqrt{K(t,t)}}$ is centered. By Assumption 1, it is Hölder continuous and $\sigma^2$-sub-Gaussian on $t \in [0, T]$. Therefore, we can directly apply Lemma 6 and get with probability at least $1 - \delta/3n$,

$$
\sup_{t \in [0,T]} \left| \frac{x_i(t) - \mu(t)}{\sqrt{K(t,t)}} \right| \leq L + \sigma \sqrt{2 \log(6Tn/\delta)}, \quad \forall i = 1, \dots, n
$$

Now by taking a union bound over $i = 1, \dots, n$, we get with probability at least $1 - \delta/3$,

$$
\sup_{i \in [1:n], t \in [0,T]} \left| \frac{x_i(t) - \mu(t)}{\sqrt{K(t,t)}} \right| \leq L + \sigma \sqrt{2 \log(6Tn/\delta)}. \tag{15}
$$

For the second term, we apply the $\varepsilon$-net and covering number argument as in Lemma 6. We discretize the time interval $[0, T]$ into $N$ parts $[0, T/N] \cup [T/N, 2T/N] \cdots \cup [(N-1)T/N, T]$, then for any $t \in [0, T]$, there exists a sub-interval such that $t \in [(k-1)T/N, kT/N]$ for some $k \in [1:N]$. Therefore, $\forall t \in [0, T]$ such that $t \in [(k-1)T/N, kT/N]$ for some $k \in [1:N]$, by Hölder continuity in Assumption 1 for the normalized process, we have

$$
\left| \sum_{i=1}^n \sigma_i \frac{x_i(t) - \mu(t)}{\sqrt{K(t,t)}} \right| \leq \left| \sum_{i=1}^n \sigma_i \frac{x_i \left( \frac{(k-1)T}{N} \right) - \mu \left( \frac{(k-1)T}{N} \right)}{\sqrt{K \left( \frac{(k-1)T}{N}, \frac{(k-1)T}{N} \right)}} \right| + \left| \sum_{i=1}^n \sigma_i \left( \frac{x_i \left( \frac{(k-1)T}{N} \right) - \mu \left( \frac{(k-1)T}{N} \right)}{\sqrt{K \left( \frac{(k-1)T}{N}, \frac{(k-1)T}{N} \right)}} - \frac{x_i(t) - \mu(t)}{\sqrt{K(t,t)}} \right) \right|
$$

$$
\leq \max_{k=1,\dots,N} \left| \sum_{i=1}^n \sigma_i \frac{x_i \left( \frac{(k-1)T}{N} \right) - \mu \left( \frac{(k-1)T}{N} \right)}{\sqrt{K \left( \frac{(k-1)T}{N}, \frac{(k-1)T}{N} \right)}} \right| + \|\sigma\| \sqrt{n} L \left( \frac{T}{N} \right)^H
$$

$$
= \max_{k=1,\dots,N} \left| \sum_{i=1}^n \sigma_i \frac{x_i \left( \frac{(k-1)T}{N} \right) - \mu \left( \frac{(k-1)T}{N} \right)}{\sqrt{K \left( \frac{(k-1)T}{N}, \frac{(k-1)T}{N} \right)}} \right| + n L \left( \frac{T}{N} \right)^H
$$

Then by the Massart Lemma 5 and the sup norm bound (15), with probability at least $1 - \delta/3$,

$$
\mathbb{E}_\sigma \left[ \sup_{t \in [0,T]} \left| \sum_{i=1}^n \sigma_i \frac{x_i(t) - \mu(t)}{\sqrt{K(t,t)}} \right| \right] \leq \mathbb{E}_\sigma \left[ \max_{k=1,\dots,N} \left| \sum_{i=1}^n \sigma_i \frac{x_i \left( \frac{(k-1)T}{N} \right) - \mu \left( \frac{(k-1)T}{N} \right)}{\sqrt{K \left( \frac{(k-1)T}{N}, \frac{(k-1)T}{N} \right)}} \right| \right] + nL \left( \frac{T}{N} \right)^H
$$

$$
\leq \sqrt{2n \log N} \cdot \sup_{i \in [1:n], t \in [0,T]} \left| \frac{x_i(t) - \mu(t)}{\sqrt{K(t,t)}} \right| + nL \left( \frac{T}{N} \right)^H
$$

$$
\leq \sqrt{2n \log N} \left( L + \sigma \sqrt{2 \log(6Tn/\delta)} \right) + nL \left( \frac{T}{N} \right)^H
$$

Since $N$ is an arbitrary integer number, we let $N = \left[ Tn^{1/H} \right] + 1$, then we get

$$
\mathbb{E}_\sigma \left[ \sup_{t \in [0,T]} \left| \sum_{i=1}^n \sigma_i \frac{x_i(t) - \mu(t)}{\sqrt{K(t,t)}} \right| \right] \leq \mathcal{O} \left( \sqrt{n \cdot \log N \cdot \log(Tn/\delta)} \right)
$$

$$
\leq \mathcal{O} \left( \sqrt{n} \log(NTn/\delta) \right) \tag{16}
$$

$$
= \mathcal{O} \left( \sqrt{n} \log(Tn/\delta) \right).
$$

Combining (15), (16), (11) and (12), we can further bound (13) as

$$
\sup_{\theta \in \Theta, i \in [1:n]} |g_\theta(x_i)| \leq \sup_{\theta \in \Theta} \left( \int_0^T |\rho_\theta(T-t)| \sqrt{K(t,t)} dt \right) \mathcal{O} \left( \sqrt{\log(Tn/\delta)} \right) + \sup_{\theta \in \Theta} \left| \int_0^T \Re(\rho_\theta(T-t)) \mu(t) dt \right| + 1 \tag{17}
$$

And the Rademacher complexity is further bounded as

$$
\mathcal{R}_S(\mathcal{F})
$$

$$
\leq \frac{2 \sup_{\theta \in \Theta, i \in [1:n]} |g_\theta(x_i)|}{n} \mathbb{E}_\sigma \left[ \sup_{\theta \in \Theta} \int_0^T \rho_\theta(T-t) \sum_{i=1}^n \sigma_i x_i(t) dt \right]
$$

$$
\leq \frac{2 \sup_{\theta \in \Theta, i \in [1:n]} |g_\theta(x_i)|}{n} \left( \sup_{\theta \in \Theta} \int_0^T |\rho_\theta(T-t)| \sqrt{K(t,t)} dt + \sup_{\theta \in \Theta} \left| \int_0^T \rho_\theta(T-t) \mu(t) dt \right| \right) \cdot \mathcal{O} \left( \sqrt{n} \log(Tn/\delta) \right)
$$

$$
\leq \left( \sup_{\theta \in \Theta} \int_0^T |\rho_\theta(T-t)| \sqrt{K(t,t)} dt + \sup_{\theta \in \Theta} \left| \int_0^T \rho_\theta(T-t) \mu(t) dt \right| + 1 \right)^2 \cdot \mathcal{O} \left( \frac{\log^{3/2}(Tn/\delta)}{\sqrt{n}} \right).
$$

Finally, by the symmetrization of $R_x(\theta) - R_n(\theta)$, combining it with (17) and (1), we have with probability at least $1 - \delta$,

$$
\sup_{\theta \in \Theta} |R_x(\theta) - R_n(\theta)| \leq \left( \sup_{\theta \in \Theta} \int_0^T |\rho_\theta(T-t)| \sqrt{K(t,t)} dt + \sup_{\theta \in \Theta} \left| \int_0^T \rho_\theta(T-t) \mu(t) dt \right| + 1 \right)^2 \cdot \mathcal{O} \left( \frac{\log^{3/2}(Tn/\delta)}{\sqrt{n}} \right).
$$

$\square$

## F    PROOF FOR PROPOSITION 1

*Proof.* First, notice that by the Hölder's inequality with $p = 1, q = \infty$, we have

$$\mathbb{E}_x \left[ \left| \int_0^T \rho_{\tilde{\theta}}(T-t)x(t)dt \right| \right]$$

$$= \frac{\mathbb{E}_x \left[ \left| \int_0^T \rho_\theta(T-t)x(t)dt \right| \right]}{\int_0^T |\rho_\theta(T-t)| \sqrt{K(t,t)}dt + \left| \int_0^T \rho_\theta(T-t)\mu(t)dt \right|}$$

$$\leq \frac{\mathbb{E}_x \left[ \left| \int_0^T \rho_\theta(T-t)(x(t)-\mu(t))dt \right| \right] + \left| \int_0^T \rho_\theta(T-t)\mu(t)dt \right|}{\int_0^T |\rho_\theta(T-t)| \sqrt{K(t,t)}dt + \left| \int_0^T \rho_\theta(T-t)\mu(t)dt \right|}$$

$$\leq \frac{\int_0^T |\rho_\theta(T-t)| \sqrt{K(t,t)}dt \cdot \mathbb{E}_x \left[ \sup_{t \in [0,T]} \left| \frac{x(t)-\mu(t)}{\sqrt{K(t,t)}} \right| \right] + \left| \int_0^T \rho_\theta(T-t)\mu(t)dt \right|}{\int_0^T |\rho_\theta(T-t)| \sqrt{K(t,t)}dt + \left| \int_0^T \rho_\theta(T-t)\mu(t)dt \right|}$$

$$\leq \mathbb{E}_x \left[ \sup_{t \in [0,T]} \left| \frac{x(t)-\mu(t)}{\sqrt{K(t,t)}} \right| \right] + 1$$

We let $X_t := \frac{x(t)-\mu(t)}{\sqrt{K(t,t)}}$, then by Assumption 1, $X_t$ is Hölder continuous and $\sigma^2$ sub-Gaussian for any $t \in [0,T]$. Again, we use an $\varepsilon$-net argument to bound $\mathbb{E} \left[ \sup_{t \in [0,T]} |X_t| \right]$. By separating the time interval $[0,T]$ into $N$ parts $[0, T/N] \cup [T/N, 2T/N] \cdots \cup [(N-1)T/N, T]$. Then for any time $t \in [0,T]$, there exists a $\pi(t) \in \{0, T/N, \ldots, (N-1)T/N\}$ such that $|t - \pi(t)| \leq T/N$. Therefore, by Hölder continuity,

$$\mathbb{E} \left[ \sup_{t \in [0,T]} |X_t| \right] \leq \mathbb{E} \left[ \sup_{t \in [0,T]} |X_t - X_{\pi(t)}| \right] + \mathbb{E} \left[ \sup_{t \in [0,T]} |X_{\pi(t)}| \right]$$

$$\leq L \left( \frac{T}{N} \right)^H + \mathbb{E} \left[ \max_{i \in [0:N-1]} |X_{iT/N}| \right].$$

For the maximum over a finite class, notice that for any $u_0 > 0$,

$$\mathbb{E} \left[ \max_{i \in [0:N-1]} |X_{iT/N}| \right] = \int_0^\infty P \left( \max_{i \in [0:N-1]} |X_{iT/N}| \geq u \right) du$$

$$= \int_0^{u_0} P \left( \max_{i \in [0:N-1]} |X_{iT/N}| \geq u \right) du + \int_{u_0}^\infty P \left( \max_{i \in [0:N-1]} |X_{iT/N}| \geq u \right) du$$

$$\leq u_0 + \int_{u_0}^\infty \sum_{i=0}^{N-1} P \left( |X_{iT/N}| \geq u \right) du.$$

Since $X_{iT/N}$ is $\sigma^2$ sub-Gaussian for every $i \in [0:N-1]$, then $\forall u_0 > 0$,

$$\mathbb{E} \left[ \max_{i \in [0:N-1]} |X_{iT/N}| \right] \leq u_0 + 2N \int_{u_0}^\infty \exp \left( -\frac{u^2}{2\sigma^2} \right) du$$

$$\leq u_0 + 2N \int_{u_0}^\infty \frac{u}{u_0} \exp \left( -\frac{u^2}{2\sigma^2} \right) du$$

$$= u_0 + \frac{2N\sigma^2}{u_0} \exp \left( -\frac{u_0^2}{2\sigma^2} \right).$$

Minimizing the above term over $u_0 > 0$, we can simply let $u_0 = \sigma\sqrt{2\log 2N}$, then

$$\mathbb{E} \left[ \max_{i \in [0:N-1]} |X_{iT/N}| \right] \leq \sigma\sqrt{2\log 2N} + \frac{\sigma}{\sqrt{2\log 2N}} \leq 2\sigma\sqrt{2\log 2N}.$$

Now back to the original upper bound, we get

$$\mathbb{E}\left[\sup_{t\in[0,T]}|X_t|\right] \le L\left(\frac{T}{N}\right)^H + 2\sigma\sqrt{2\log 2N}.$$

Since $N$ is an arbitrary positive integer, we simply take $N = [T] + 1$, finally we get

$$\mathbb{E}\left[\sup_{t\in[0,T]}|X_t|\right] \le L + 2\sigma\sqrt{2\log(2T+2)} = \mathcal{O}(\sqrt{\log T}).$$

## G  LIPSCHITZ FUNCTION OF SUB-GAUSSIAN RANDOM VARIABLES

In this section, we provide some known examples for the sub-Gaussian random variables that remain the sub-Gaussian property under a Lipschitz function.

1. For a bounded random variable $X \in [0, 1]$, if $f : [0, 1] \to \mathbb{R}$ is a quasi-convex function, i.e., $\{x : f(x) \le s\}$ is a convex set for all $s \in \mathbb{R}$. If $f$ is further Lipschitz in $[0, 1]$, then $f(X)$ is sub-Gaussian. See Theorem 7.12 in Boucheron et al. (2013).

2. For a sub-Gaussian random variable $X$ that has density of the form $e^{-U(x)}$ with $U$ being twice continuously differentiable and $U''(x) > 0, \forall x \in \mathbb{R}$, then if $f$ is a Lipschitz function, $f(X)$ is also sub-Gaussian. See Theorem 5.2.15 in Vershynin (2020).

$\square$

