# OpenReview forum: "From generalization analysis to optimization designs for state space models"
_ICLR.cc/2024/Conference — Submitted to ICLR 2024_

### Official Review · Reviewer_SPHH · 2023-10-18

**Soundness:** 3 good
**Presentation:** 3 good
**Contribution:** 2 fair
**Rating:** 6
**Confidence:** 4

**Summary:**

This paper proves a generalization error bound for SSMs, where the input data are assumed to be sampled from a Gaussian process, which incorporates temporal dependency. The error bound motivates both a new initialization scaling strategy and a regularized loss function in training. The effect of the new initialization and regularization are evaluated using both a synthetic dataset and the Long-Range Arena benchmark collection.

**Strengths:**

* The error bound in Theorem 1 incorporates the temporal dependency of the input data. It is nicely justified in the `Comparison` section why this is important and how it is missing from the previous work.
* As far as I know, the scaling and the regularization strategies are new to the SSM society. The paper demonstrates their potential promises using experiments.
* Overall, the paper is clearly written, and the mathematical statements are mostly properly made. (See `Questions` below for a couple of clarification questions.)

**Weaknesses:**

* My biggest concern is the assumption that the inputs in `Theorem 1` are sampled from a Gaussian process. Perhaps the Gaussian assumption is more reasonable in a non-temporal setting, such as linear regression. However, most time series inputs that we encounter in practice cannot be ''sampled'' from a Gaussian process. For example, you cannot find a single GP that accounts for the flattened MNIST images, because images representing different numbers may have their own unique features, and such distinct features cannot be captured solely by the randomness in your GP. If you fix a GP and randomly sample your sequential pixels, then most figures you obtain won't represent any number. I understand that this Gaussian assumption is crucial in proving your generalization error bound and there is perhaps no way out, but this indeed results in a gap between your theory and the methodologies you proposed.

* The proposed regularization method (9) combines a normalized $\ell^2$ loss and a regularization term. In an SSM, one usually chains multiple LTI systems; however, only the target output of the entire SSM is known (e.g., whether the maze is solvable, what the number in the MNIST figure is). In that case, it is unclear how the ''target outputs'' $y_i$ of the intermediate LTI systems are defined.

* The evaluation of the model does not show clear evidence of why the scaling of the initialization makes the model more robust. For example, in `Table 1`, comparing the cases `w/o (8), (9)` to `w/ (8)`, it seems that adding the scaling improves the training accuracy but makes the generalization accuracy even worse. This actually contradicts the claim that the model is made more robust by scaling the initialization.

* Since the regularization involves a hyperparameter $\lambda$, it is a good practice to perform an ablation study to demonstrate the effect of changing $\lambda$.

**Questions:**

* The setup of this paper does not consider the matrix $\mathbf{D}$ in an LTI system. How easy is it to incorporate that matrix and do you need to scale $\mathbf{D}$ in initialization?

* In `Theorem 1`, can you show the explicit dependency of $C_T$ on $T$? This is important because in training an SSM, the discretization size $\Delta$ is usually trainable, making the final time $T$ in the continuous world change from time to time. Hence, in order to apply your theory, it is better if we can understand the role of $T$.

* In `Theorem 1`, when you say $\tilde{\mathcal{O}}(\cdot)$ hides the logarithmic factor, which variables are considered? For example, it clearly does not hide $\log(1/\delta)$.

* The presence of `Proposition 1` seems a bit abrupt. How does that relate to your ''robustness'' discussion? In addition, what kind of ''stability'' are you referring to? This is a fairly ambiguous term, which can represent the stability of a numerical algorithm, the asymptotic stability of your LTI system (i.e., if your eigenvalues of $\mathbf{A}$ are all in the left half-plane), or something else.

* In your experiments, it is shown that the regularizer improves the training accuracy, which is a bit counter-intuitive. Do you have a justification for that?

* Not a question but a side note: in order to comply with the ICLR formatting guide, all matrices and vectors should be made boldface.

---

> ### Author Response · Authors · 2023-11-22
> **Authors' response**
>
> We thank the reviewer for the valuable comments and suggestions. In the following, we address the 4 weaknesses from Weakness 1 to Weakness 4 and the 5 questions from Q1 to Q5.
>
> Weakness 1. *My biggest concern is the assumption that the inputs in Theorem 1 are sampled from a Gaussian process.*
> * We agree with the reviewer that the Gaussian process assumption is a strong assumption. To address this concern, we have relaxed the Gaussian process assumption to a more general class of random processes that are sub-Gaussian and Hölder continuous.
> This is a more general assumption than the Gaussian process assumption in the sense that it covers a larger class of random processes including Gaussian processes, Hölder continuous bounded processes.
> In particular, as we mentioned in the first point of the response *To all reviewers*, when the range of the random process is a finite set, then the Hölder continuity condition can be dropped.
> This relaxation includes the case of image classification tasks, where the input image is flattened as a sequence and the range for each pixel value is a finite set, e.g., $\{0,1,\ldots,255\}$.
> **In the revised paper, we have substantially improved the generality of our theorem (see the revised statement in Theorem 1 and proof in Appendix E) by dropping the Gaussian process assumption and replacing it with a much weaker one in Assumption 1.**
>
> Weakness 2. *The proposed regularization method (9) combines a normalized $\ell^2$ loss and a regularization term. In an SSM, one usually chains multiple LTI systems; however, only the target output of the entire SSM is known (e.g., whether the maze is solvable, what the number in the MNIST figure is). In that case, it is unclear how the ''target outputs'' $y_i$ of the intermediate LTI systems are defined.*
> * We would like to mention that the regularized loss by the regularization method (9) is given by the original loss plus the regularization measure $\tau(\theta)$.
> When training multi-layer SSMs, we calculate the complexity $\tau(\theta)$ at each layer and add them together as a total regularization measure.
> Notice that the complexity for each single layer only depends on the input sequence statistics and the model parameters of that layer.
> Therefore, we do not need to know how the target outputs $y_i$ of the intermediate systems are defined.
> **To make the statement more clear, we have changed the notation for the regularized loss (9) to $\tilde{R}_n(\theta) := R_n(\theta) +  \lambda  \cdot \tau (\theta)$ and added the above explanations in Section 4.3.**
>
> Weakness 3. *The evaluation of the model does not show clear evidence of why the scaling of the initialization makes the model more robust. For example, in Table 1, comparing the cases w/o (8), (9) to w/ (8), it seems that adding the scaling improves the training accuracy but makes the generalization accuracy even worse. This actually contradicts the claim that the model is made more robust by scaling the initialization.*
> * There is a misunderstanding here due to the ambiguity of the term `robustness`.
> We would like to emphasize that 'robustness' in the paper refers to the robustness of the *output value scales* (instead of the test performance) on SSMs across different temporal dependencies of the input sequence.
> This is desirable because one does not a priori have a good sense of the input sequence and the temporal dependency of the input sequence may vary a lot.
> Therefore, a good way to initialize SSMs should be one that is not sensitive to the variations of the temporal structures.
> This is not true for the original HiPPO based initialization scheme, which is independent of the input sequence.
> By looking at the left-hand side of Figure 2, we can see that the SSM output scales at initialization without rescaling (dashed lines) are very sensitive to the variations of temporal dependencies, while the SSM output scales at initialization with rescaling  (solid lines) are much more robust.
> In Table 1, we can also see that adding the scaling improves the training performance but the generalization performance is not guaranteed to be improved.
> The above empirical findings also match Proposition 1, which shows that the scale of the SSM output scales after rescaling does not depend on the variations of the temporal structures.
> **To make the statement more clear, we have added the above discussions in Section 4.4.**

---

> > ### Author Response · Authors · 2023-11-22
> > **Authors' response (continue)**
> >
> > Weakness 4. *Since the regularization involves a hyperparameter $\lambda$, it is a good practice to perform an ablation study to demonstrate the effect of changing $\lambda$.*
> > * **We have added more numerical results on the sensitivity of the regularization coefficient $\lambda$ for unidirectional S4-Legs, bidirectional S4-Legs, bidirectional S4D-Legs in Section A.2**.
> > The results in **Table 6, 7, 8** show that the test performance is much more sensitive to $\lambda$ for the Pathfinder and PathX tasks compared to other tasks.
> > This is because the generalization measure for the Pathfinder and PathX tasks are much larger than other tasks (as shown in **Figure 3, 4, and 5**), making the generalization measure very sensitive to the magnitude of $\lambda$.
> > The ablations study on the sensitivity of $\lambda$ mathches the intuition that the Pathfinder and PathX tasks are more challenging than the other tasks.
> >
> > Q1. *The setup of this paper does not consider the matrix $D$ in an LTI system. How easy is it to incorporate that matrix and do you need to scale $D$ in initialization?*
> > * As we mentioned in the third point of the response *To all reviewers*, our generalization bound can be extended to the case with skip connection $D$.
> > This is because that our generalization bound does not have a restriction on the kernel function $\rho_\theta(s)$. When adding $D$, we can incorporate it into the kernel function to get a new kernel $\rho_\theta(s) + D \delta (s)$, where $\delta(\cdot)$ is a delta function.
> > However, the initialization scheme may not work well when $D x(t)$ is much larger than $\int_0^t \rho_\theta(s) x(t-s)$.
> > This is because that the initialization scheme only adjusts $C$ and requires the kernel function to be linear in $C$.
> > In that case, $D$ should also be rescaled.
> > **We have added the above discussions in Section 5.**
> >
> > Q2. *In Theorem 1, can you show the explicit dependency of $C_T$ on $T$?*
> > * Yes, we have modified the generalization bound in Theorem 1 to show the explicit dependency of $C_T$ on $T$, which is given by $\sqrt{\log T}$.
> >
> > Q3. *In Theorem 1, when you say $\tilde{\mathcal{O}}(\cdot)$ hides the logarithmic factor, which variables are considered?*
> > * In light of the question, we use the notation $\mathcal{O}$ instead of $\tilde{\mathcal{O}}$ in Theorem 1, where $\mathcal{O}$ hides a constant that depends on $\sigma, L, H$ (defined in Assumption 1).
> > **We have modified the notation in Theorem 1 to make it more clear.**
> >
> > Q4. *The presence of Proposition 1 seems a bit abrupt. How does that relate to your ''robustness'' discussion? In addition, what kind of ''stability'' are you referring to? This is a fairly ambiguous term, which can represent the stability of a numerical algorithm, the asymptotic stability of your LTI system*
> > * In view of the comment, we have changed 'robustness' to 'robustness of the output value scales on SSMs' in the revised paper.
> > The 'stability' in the original paper refers to the stability of the training loss value.
> > **To make the statement more clear, we add the above explanations after Section 4.2.**
> >
> > Q5. *In your experiments, it is shown that the regularizer improves the training accuracy, which is a bit counter-intuitive. Do you have a justification for that?*
> > * Maybe there is a misunderstanding here. Our experiments do not show that the regularizer improves the training accuracy. By looking at Table 1, when comparing w/o (8, 9) vs w (9) and w (8) vs w (8, 9), we can see that the final training loss is not improved by adding the regularizer (9).
> > However, with the initialization scheme (8), the training performance is improved.
> > This is shown in Table 1 by comparing w/o (8, 9) vs w (8) and w (9) vs w (8, 9).

---

### Official Review · Reviewer_nrdv · 2023-10-29

**Soundness:** 3 good
**Presentation:** 3 good
**Contribution:** 3 good
**Rating:** 6
**Confidence:** 4

**Summary:**

This paper investigates the generalization performance of state space model, in which the data-dependent generalization bound are established. Motivated by the the theoretical findings, the authors design a scaling rule for model initialization and introduce a new regularization mechanism to improve both the robustness and generalization performance of SSM.

**Strengths:**

1. Each section of the paper is clear presented and motivates the paper well.
2. The generalization results appear to be interesting, and the experimental results support the theoretical claims.

**Weaknesses:**

The current theoretical results could be more plentiful, e.g. replenish the generalization analysis on  regularized model (9), which may help to answer the question raised below.

**Questions:**

1. In Theorem 1 , the authors claim that the SSM generalization is characterized by the temporal dependencies of the sequential data. More details on how does the dependency of the sequential data affect the generalization error should be included. Moreover, in order to achieve small generalization error, the mean and variance of the GP should remain a small level. While these two key parameters rely on the GP assumption, independent  of data. This seems inconsistent with data-dependent generalization error bounds, as claimed in the paper.

2. In speak of enhancing the robustness of SSMs on different temporal dependencies, the authors take   $1/\sqrt{\tau(\theta)}$  as a rescaling factor for initialization. Any theoretical guarantees (e.g. variance analysis)  on the robustness comparing with the HiPPO framework?

3. The main techniques adopted in the proof are sub-exponential property of r.v. and Borell-TIS inequality, how did they yield to  temporal dependency generalization bounds since both of them are temporal independent.

---

> ### Author Response · Authors · 2023-11-22
> **Authors' response**
>
> We thank the reviewer for the valuable comments and suggestions. In the following, we address questions from Q1 to Q3.
>
> Q1. *In Theorem 1 , the authors claim that the SSM generalization is characterized by the temporal dependencies of the sequential data. More details on how does the dependency of the sequential data affect the generalization error should be included. Moreover, in order to achieve small generalization error, the mean and variance of the GP should remain a small level. While these two key parameters rely on the GP assumption, independent of data. This seems inconsistent with data-dependent generalization error bounds, as claimed in the paper.*
> * Our generalization bound is data-dependent in the sense that it depends on the statistics of the input sequence and we can only calculate the bound after observing the input sequence.
> In particular, we can see that the dominant term is given by
> $$
> \psi(\theta) := \int_0^T \left|\rho_{\theta} (T-s)\right| \sqrt{K(s, s)}ds +\left|\int_0^T  \rho_\theta(T-s) \mu(s) d s\right|.
> $$
> For the SSM basis function $\rho_\theta(s) = C e^{As}B$, it is exponentially decay with respect to $s$ because the eigenvalues of $A$ are negative.
> Therefore, we do not require the mean and variance to be uniformly small in $[0, T]$ to get a small generalization gap.
> We also relax the GP assumption to a more general class of random processes that are sub-Gaussian and Hölder continuous, and we refer the reviewer to the response to the comment 5 from Reviewer 2971 for more details.
> **We have added the above discussions to make the generalization bound more clear after Theorem 1.**
>
> Q2. *In speak of enhancing the robustness of SSMs on different temporal dependencies, the authors take $1/\sqrt{\tau(\theta)}$
>  as a rescaling factor for initialization. Any theoretical guarantees (e.g. variance analysis) on the robustness comparing with the HiPPO framework?*
>  * We would like to emphasize that the rescaling method is built on the HiPPO based initialization, and it is applied before training.
>  The HiPPO based initialization method is independent of the input sequence while we rescale the initialization based on the input sequence.
>  This rescaling method improves the robustness of the initial
>  output value scales of SSMs across different temporal dependencies of the input sequence.
>  As Proposition 1 shows, the scale of the SSM output values after rescaling does not depend on the variations of the temporal structures.
>  **To make the statement more clear, we have added the above discussions in Section 4.2.**
>
>  Q3. *The main techniques adopted in the proof are sub-exponential property of r.v. and Borell-TIS inequality, how did they yield to temporal dependency generalization bounds since both of them are temporal independent.*
>  * We would like to emphasize that the sub-Gaussian property and the Hölder continuity condition in Assumption 1 are applied to sequences in a time interval $[0, T]$ instead of a single time point (see equation (14) for the terms that we need to bound) in our proof of Theorem 1. This is where the temporal dependency comes from.
>  Combining the $\varepsilon$-net argument and the covering number tools, we can apply the sub-Gaussian property to a sequence in a time interval $[0, T]$.
>  Since we have relaxed the Gaussian process assumption to a more general class of random processes that are sub-Gaussian and Hölder continuous, the proof of Theorem 1 has also been revised accordingly.
>  **In light of the question, we added more details in the proof sketch of Theorem 1.**

---

### Official Review · Reviewer_2971 · 2023-10-31

**Soundness:** 2 fair
**Presentation:** 2 fair
**Contribution:** 2 fair
**Rating:** 6
**Confidence:** 3

**Summary:**

This paper presents an analysis of a generalization bound for linear SSMs.  These SSMs are the building blocks of a new class of deep sequence models.  The authors posit that understanding this bound promises to inform the design of initialization and regularization schemes.  Networks trained using these techniques are shown to have better performance or favorable training characteristics on two simple examples.

**Strengths:**

Studying the scaling properties of different initialization schemes and regularizers is clearly important.  Other papers have started to study this also in a bid to understand if HiPPO is simply an example of a wider family of options.  Furthermore, regularizers arising naturally from generalization (which is what a regularizer is trying to target!) is intuitively appealing.  The detail the authors go through their derivations is (for better or for worse) incredible.  I have not gone through the derivations line by line, but I think I understand the general gist of them.  The intuitions given are enough to allow most readers to grasp the core concepts.  The experimental results show promise.  Overall, the paper is fairly well written and prepared.

**Weaknesses:**

I am very on the fence on this work.  I think the work is sound, and the authors are to be commended for the detail they go into, but I am not quite convinced that it is at the requisite threshold for acceptance.  Ironically, I am actually left wanting slightly more.  As someone who uses SSM models, I am not yet convinced to integrate this into our workflow, and would need to see more evidence that it is worth incorporating.  Furthermore, I think there are some disconnects between the theory and the practice.

My main comment is that the experimental evaluation isn’t quite complete enough to convince me:
- There is quite a lot of work here to just get better generalization as shown on a near-pathological synthetic example, and marginal improvement on LRA (see Q.1. as well).
- I would have also liked to have seen more evaluation of the initialization across different sizes of models, sensitivity to hyperparameters etc.  Experimental repeats are also important to ensure that the results are reliable.
- The additional time complexity is also theoretically commensurate with the original S4 model, but I would like to see a concrete comparison of the runtimes to confirm this.
- I would also like to see a more thorough comparison to, e.g., the initialization and metrics suggested by Orvieto et al. [2023], or evaluation of whether this initialization/regularization scheme can be applied to methods adjacent to S4 (e.g. S4D, S5, Liquid-S4, MEGA).
- How reasonable are the assumptions, and how tight are the bounds in practice?  I do not have a great understanding of whether the GP assumption is sensible in practice, and there doesn’t appear to be any validation of this.  How does the fidelity of the GP approximation impact the performance of the regularizer?
- It would be interesting to try and establish exactly how the initialization and regularization terms affect the learned model.  I understand that L2 regularization reduces the magnitude of the parameters (controlling a notion of complexity), but what does the regularizer in (9) actually encourage in the learned models?  How are the regularized models different from regular S4 models?  This analysis might enable the design of even better SSM structures.
- It is also a shame that Path-X wasn’t included in the paper.  My understanding is that Path-X is the only really challenging LRA benchmark.  While I am willing to overlook this in this evaluation, I encourage the authors to complete the benchmarks.

There are additional results in the supplement that are basically not commented on, and seem important (e.g. Figure 3 and Figure 4).  These should be explained more thoroughly, and brought up to the main if they are truly important.  I think these extra experiments that probe the method are super important to verifying that the method is working as expected.

**Summary**:  Right now I am just about convinced that the method just about works, but I think some arguments and opportunities aren’t fully explored.  There is clearly an opportunity for this line of work to become very impactful, but I think it would benefit from a round of revisions, and expanding the breadth and depth of the experimental evaluation.  That said, I am very open to revising my review score should the authors remedy some of my concerns.

## Minor comments
- Figures, tables etc should be floated to the top or bottom of pages, as opposed to inserted in the middle of the text.
- Table 1 should be prepared using the same (and correct) style as used by Table 2.
- Only proper nouns should be capitalized.

**Questions:**

**Q.1.**:  Can the authors clarify whether, in Table 2, w/o (8, 9) corresponds to the original S4 model?  The numbers are slightly lower than in the original paper, and I am trying to clarify whether these numbers are like-for-like within the table, and how comparable to S4 they are.

**Q.2.**:  The theories and algorithms presented are for one-layer networks, but then in Section 4.4 you use multilayer networks.  Can the authors comment how the theories translate to multi-layer networks, where, presumably, the statistics of the input to each layer are not constant.

**Q.3.**:  Can you clarify how the rescaling in Line 7 of Algorithm 1 works: (a) across epochs and (b) extends to multiple layers.  R.e. (a): is the value of $\tilde{C}$ rescaled at the beginning of every epoch?  I.e. it is being “reinitialized” by rescaling its previous values.  R.e. (b): does rescaling $\tilde{C}$ at each layer disrupt the action of other layers?  Or is there a different method for rescaling between layers?

**Q.4.**:  The experiment in Figure 2, is it really Gaussian white noise?  Or is it more like Brownian motion?

**Q.5.**:  Does the training loss in Figure 2 (right) include the regularization term?  I believe it should actually be labeled as “Training set MSE”.

**Q.6.**:  An appealing benefit of S4 is the zero-shot transfer to other sampling frequencies.  However, this might change the scale of the time-dependencies.  Can the authors clarify whether there are drawbacks to this method with respect to zero-shot transfer?

**Q.7.**:  The theory is presented for linear SSMs, but in practice, multi-layer S4 models are interleaved with position-wise nonlinearities.  I cannot see any discussion of how these nonlinearities (and the parameters in these nonlinearities, e.g. GLU) interact with the regularization of the parameters in the SSM, or, how the warping effect of the nonlinearity affects/interacts with the theoretical results.

---

> ### Author Response · Authors · 2023-11-22
> **Author's response**
>
> We thank the reviewer for the valuable comments and suggestions. In the following, we address the comments for the 7 comments and questions from Q1 to Q7.
>
>  Comment 1. *There is quite a lot of work here to just get better generalization as shown on a near-pathological synthetic example, and marginal improvement on LRA (see Q.1. as well).*
>
>  Q1. *Can the authors clarify whether, in Table 2, w/o (8, 9) corresponds to the original S4 model? The numbers are slightly lower than in the original paper, and I am trying to clarify whether these numbers are like-for-like within the table, and how comparable to S4 they are.*
>   * We would like to emphasize that this paper is not just about getting better generalization. The theory is interesting by itself as it is the first data-dependent generalization bound for SSMs that is based on the temporal dependency of the input sequence.
>   For the improvement on LRA, **we have added more numerical results for bidirectional models such as bidirectional S4-Legs and bidirectional S4D-Legs in Table 2 and Appendix A.2.**
>   The results in the updated Table 2 show that our proposed regularization method consistently improves the generalization performance in most tasks for all three models.
>   In that sense, the improvement on LRA is not marginal.
>   * Our original version used the unidirectional S4-Legs model (Gu 2022a), which processes a sequence in one direction from past to future.
>   Compared to recent commonly used bidirectional models (Gu 2023), the unidirectional model is closest to our problem setting even though it is not the state-of-the-art model.
>   Therefore, the numbers in Table 2 should be compared with the results in the earliest paper (Gu 2022a), and we can see that our numbers match with the results in (Gu 2022a).
>
>  Comment 2. *I would have also liked to have seen more evaluation of the initialization across different sizes of models, sensitivity to hyperparameters etc. Experimental repeats are also important to ensure that the results are reliable.*
>  * **We have added more numerical results on the sensitivity of the regularization coefficient $\lambda$ for unidirectional S4-Legs, bidirectional S4-Legs and bidirectional S4D-Legs in Section A.2**.
>  The results in **Table 6, 7, 8** show that the test performance is much more sensitive to $\lambda$ for the Pathfinder and PathX tasks compared to other tasks.
>  This is because the generalization measure for the Pathfinder and PathX tasks are much larger than other tasks (as shown in **Figure 3, 4, and 5**), making the generalization measure very sensitive to the magnitude of $\lambda$.
>  The ablations study on the sensitivity of $\lambda$ matches the intuition that the Pathfinder and PathX tasks are more challenging than the other tasks.
>  For the repeated experiments, we have added the standard deviation (in 3 independent runs) of the test accuracy for the PathX task in **Table 2** for the bidirectional models.
>
> Comment 3. *The additional time complexity is also theoretically commensurate with the original S4 model, but I would like to see a concrete comparison of the runtimes to confirm this.*
> * **We have added the running time per epoch in the updated Table 2 in Section 4.4.**
> The running time comparison shows that the proposed regularization method brings a little extra computational cost compared to the original training method.
>
> Comment 4. *I would also like to see a more thorough comparison to, e.g., the initialization and metrics suggested by Orvieto et al. [2023], or evaluation of whether this initialization/regularization scheme can be applied to methods adjacent to S4 (e.g. S4D, S5, Liquid-S4, MEGA).*
> * **We have added more evaluation on the bidirectional S4D-Legs model in Table 2 and Appendix A.2.**
> The results in **Table 2** show that our proposed regularization method and initialization scheme can improve the generalization performance.

---

> > ### Author Response · Authors · 2023-11-22
> > **Author's response (continue)**
> >
> > Comment 5. *How reasonable are the assumptions, and how tight are the bounds in practice? I do not have a great understanding of whether the GP assumption is sensible in practice, and there doesn’t appear to be any validation of this. How does the fidelity of the GP approximation impact the performance of the regularizer?*
> > * We have relaxed the Gaussian process assumption to a more general class of random processes that are sub-Gaussian and Hölder continuous.
> > This is a more general assumption than the Gaussian process assumption in the sense that it covers a larger class of random processes including Gaussian processes, Hölder continuous bounded processes.
> > In particular, as we mentioned in the first point of the response *To all reviewers*, when the range of the random process is a finite set, then the Hölder continuity condition can be dropped.
> > This relaxation includes the case of image classification tasks, where the input image is flattened as a sequence and the range for each pixel value is a finite set, e.g., $\{0,1,\ldots,255\}$.
> > **In line with the comment, we provide more discussions of the new assumption in Section 4.1 and Appendix C, and modified the proof of Theorem 1 accordingly.**
> >
> > Comment 6. *It would be interesting to try and establish exactly how the initialization and regularization terms affect the learned model. I understand that L2 regularization reduces the magnitude of the parameters (controlling a notion of complexity), but what does the regularizer in (9) actually encourage in the learned models? How are the regularized models different from regular S4 models? This analysis might enable the design of even better SSM structures.*
> > * The regularization method in the paper is different from weight decay which tries to minimize the magnitude of the parameters, but to minimize the correlation between the statistics of the input sequence and the SSM basis function.
> > As we emphasize in the introduction part of the paper, regular S4 models do not regularize the hidden state matrix $A$ because its imaginary part controls the oscillating frequencies of the SSM basis function (Gu 2022b).
> > Instead of penalizing the weight norm, our regularization method regularizes the magnitude of the SSM output value, which is determined by the correlation between the input sequence statistics and the SSM basis function.
> > * For the initialization scheme, our proposed method is built on the original HiPPO based initialization, which is a *data-independent* initialization method.
> > However, when the temporal dependency of the input sequence varies, the SSM output value may vary a lot (see the dashed lines in Figure 2).
> > With our proposed initialization scheme, the SSM is more robust in the sense that the scale of the SSM output value does not depend on the variations of the temporal structures.
> > **We have added the above discussions in Section 4.2.**
> >
> > Comment 7. *It is also a shame that Path-X wasn’t included in the paper. My understanding is that Path-X is the only really challenging LRA benchmark. While I am willing to overlook this in this evaluation, I encourage the authors to complete the benchmarks.*
> > * **In line with the comment, we complete the PathX task for both directional S4-Legs and directional S4D-Legs models in Section 4.4 and Appendix A.2.**
> > The results for the PathX task show that our proposed optimization designs improve the generalization performance for both directional models.
> >
> > Minor comments for figures and tables are fixed in the revised paper.
> > Below is the response to the questions from Q2 to Q7 (Q1 is answered above in the comment 1).
> >
> > Q2. *The theories and algorithms presented are for one-layer networks, but then in Section 4.4 you use multilayer networks. Can the authors comment how the theories translate to multi-layer networks, where, presumably, the statistics of the input to each layer are not constant.*
> > * We agree with the reviewer that our theory can only work for single-layer SSMs, which is a limitation in the current paper.
> > Even in this simple case, the generalization measure is a new thing, we feel it is important to first understand the generalization of SSMs in the simplest setting.
> > For multi-layer SSMs, our algorithm simply applies the proposed regularization method to each single layer and adds the regularization measure together.
> > To extend the theory to multi-layer SSMs, we think it is an interesting direction to explore and leave it for future work.
> > **We have added the above discussions in Section 5.**

---

> ### Author Response · Authors · 2023-11-22
> **Author's response (continue)**
>
> Q3. *Can you clarify how the rescaling in Line 7 of Algorithm 1 works: (a) across epochs and (b) extends to multiple layers. R.e. (a): is the value of $\tilde{C}$
> rescaled at the beginning of every epoch? I.e. it is being “reinitialized” by rescaling its previous values. R.e. (b): does rescaling $\tilde{C}$ at each layer disrupt the action of other layers? Or is there a different method for rescaling between layers?*
> * The rescaling method is only applied for SSMs before training. We calculate the complexity measure layer by layer and then rescale the matrix $C$ from the first layer to the last layer sequentially.
> For example, for a $k$-layer SSM with the initial matrix $C_1,\ldots,C_k$ at each layer, we first calculate the complexity measure $\tau_1(\theta)$ for the first layer and rescale $C_1$ by $C_1/\sqrt{\tau_1(\theta)}$.
> Then we calculate the complexity measure $\tau_2(\theta)$ for the second layer for the updated input sequence of layer 2 and rescale $C_2$ by $C_2/\sqrt{\tau_2(\theta)}$. We repeat this process until the last layer.
> **We have added the above discussions in Section 4.2.**
>
> Q4. *The experiment in Figure 2, is it really Gaussian white noise? Or is it more like Brownian motion?*
> * It is Gaussian white noise. The autocovariance function for Brownian motion is given by $K(s,t) = \min(s,t)$ rather than $\frac{1}{|b| \sqrt{\pi}} e^{-((s-t)/b)^2}$.
>
> Q5. *Does the training loss in Figure 2 (right) include the regularization term? I believe it should actually be labeled as “Training set MSE”.*
> * Figure 2 (right) shows the training loss when training with the initialization scheme but without the regularization method.
> **We have highlighted it in Section 4.2.**
>
> Q6. *An appealing benefit of S4 is the zero-shot transfer to other sampling frequencies. However, this might change the scale of the time-dependencies. Can the authors clarify whether there are drawbacks to this method with respect to zero-shot transfer?*
> * This is an interesting question. In fact, our generalization measure is zero shot transerable because the input sequence statistics $\sqrt{K(s,s)}, \mu(s)$ in our generalization bound both have the same scaling as the input sequence $x(s)$, i.e., by changing sequence $x(s)$ to $x(k s)$ for $k>0$, the standard deviation and the mean of $x(ks)$ is changed to $\sqrt{K(ks,ks)}, \mu(ks)$, respectively.
> Then the generalization measure is invariant after changing the sampling frequency if the SSM output value is also invariant.
> Therefore, the zero shot transferability for SSMs is automatically inherited by our generalization bound.
> **In line with the question, we have added a thorough discussion at the end of Section 4.1.**
>
> Q7. *The theory is presented for linear SSMs, but in practice, multi-layer S4 models are interleaved with position-wise nonlinearities. I cannot see any discussion of how these nonlinearities (and the parameters in these nonlinearities, e.g. GLU) interact with the regularization of the parameters in the SSM, or, how the warping effect of the nonlinearity affects/interacts with the theoretical results.*
> * Since we have relaxed the original assumption of Gaussian process to a more general class of random processes that are sub-Gaussian and Hölder continuous, the theory can be extended if the nonlinearities do not affect the Hölder condition and the sub-Gaussian property.
> In fact, the assumption is to ensure that the input sequence is bounded (with high probability) in a given time interval.
> Therefore, intuitively, if the nonlinearity function has a controllable growth rate, then the boundness of the input sequence with the nonlinearity is still satisfied and our theory works if one considers a single layer SSM.
> For example, for Lipschitz nonlinearities, we list some examples in **Appendix C** to show that the Hölder condition and the sub-Gaussian property are still satisfied.
> **We have added the above discussions in Section 5.**

---

### Author Response · Authors · 2023-11-22
**To all reviewers**

We thank all reviewers for their valuable comments and suggestions.
We have revised the paper with the main changes in blue according to the comments and suggestions.
We also provide a detailed response to each comment below.

In the following, we address three main issues raised by the reviewers:
 1. *The Gaussian process assumption in this paper is not well justified.*
 We initially focused on Gaussian process inputs as they are natural settings for certain continuous time series
 prediction problems.
 In view of the reviewers' comments, we have relaxed the Gaussian process assumption to a more general class of random processes that are sub-Gaussian and Hölder continuous in time (sequence index).
 Specifically, the sub-Gaussian assumption is used to ensure the boundness of the random process in a finite time set, and the Hölder continuity condition guarantees that the boundness of the random process can be extended to the whole (infinite) time set.
 If the range of the random process is a finite set, then the Hölder continuity condition is not needed.
 For example, in image classification tasks, when the input image is flattened as a sequence, the range for each pixel value is always a finite set, e.g., $\{0,1,\ldots,255\}$.
 In that case, the Hölder continuity condition can be dropped and the sub-Gaussian assumption is satisfied since the magnitudes of image sequences are bounded.
 **In the revised paper, we have substantially improved the generality of our theorem (see the revised statement in Theorem 1 and proof in Appendix E) by dropping the
 Gaussian process assumption and replacing it with a much weaker one in Assumption 1. Our results now apply much more generally, including to the case suggested by Reviewer SPHH, and this also addresses the comment from Reviewer 2971.**

 2. *The numerical evaluation is not convincing.*
    To address this concern, **we have added more numerical results in Section 4.4 and Appendix A.2.**
    Specifically, we have added the results of the proposed methods on two more models: bidirectional S4-Legs and bidirectional S4D-Legs.
    The original model S4-Legs used in this paper is unidirectional, which is closest to our problem setting but is not the state-of-the-art model.
    The test accuracy on the LRA benchmark in Table 2 shows that our proposed regularization method improves the generalization performance in most tasks for all three models.
    The plots of the generalization measure across the three models in Figure 3, 4, and 5 validate that our proposed generalization bound captures the generalization performance in the sense that bidirectional S4-Legs has the smallest generalization measure and best generalization performance among the three models.
    In addition, we also added ablation studies on the regularization parameter $\lambda$ in Appendix A.2.
    The results show that the Pathfinder and the PathX tasks are more sensitive to $\lambda$ than the other tasks.
    This is consistent with the intuition that the Pathfinder and the PathX tasks are more challenging than the other tasks.

 3. *There are some gaps between the theory and the methodologies.*
 The first gap is that the skip connection matrix $D$ is omitted in our theory.
 To address this concern, **we have added more discussions in Appendix D.**
 Specifically, our generalization bound can be extended to the case with skip connections.
 The reason is that by adding the matrix $D$, the expression for the SSM becomes $y(t) = \int_0^t (\rho_\theta(s) + D \delta (s)) x(t-s) d s$ where $\delta(\cdot)$ is a delta function, which is still a convolution model with a new kernel $\rho_\theta(s) + D \delta (s)$.
 Thus, we can also get a generalization bound by simply plugging this new kernel into the generalization bound in Theorem 1 because it does not have a restriction on the kernel function.
 The second gap is that the theory is only for single-layer linear SSMs, but the proposed regularization method is for multi-layer SSMs with nonlinearities.
 In fact, our generalization bound still works if the nonlinearity does not affect the Hölder condition and the sub-Gaussian property (as in Assumption 1) of the random process.
 For Lipschitz nonlinearities, we list some examples in **Appendix C** to show that the Hölder condition and the sub-Gaussian property are still satisfied.
 For multi-layer SSMs, our algorithm simply applies the proposed regularization method to each single layer and adds the regularization measure together.
 This is currently empirical, and it is an interesting direction to extend our theory to multi-layer SSMs, which is out of the scope of the current paper.
 **The above discussions have been added in Section 5.**

---

### Meta-Review · Area_Chair_tMjL · 2023-12-09

**Metareview:**

The paper investigates the generalization performance of SSMs offering theoretical analysis and practical optimization strategies. The authors propose a data-dependent generalization bound for SSMs, highlighting the interaction between SSM parameters and temporal dependencies in training sequences.

Strengths: The paper presents a novel data-dependent generalization bound for SSMs.
Practical Optimization Strategies: The proposed scaling rule and regularization method are practical, addressing real-world challenges in training SSMs. The methodologies proposed improve the robustness of SSMs to different temporal patterns in sequence data.

Weaknesses:
- The original Gaussian process assumption was a significant limitation, though it was later relaxed to a more general sub-Gaussian and Hölder continuous process.
- There's a noted disconnection between the theoretical analysis and practical application, particularly in multi-layer SSMs with nonlinearities.
- Some reviewers expressed concerns about the comprehensiveness of the empirical evaluations, suggesting more extensive testing could strengthen the paper.

During the AC/Reviewers discussion period, the reviewers brought up concerns about the paper not yet being ready for publication, mainly due to the rushed empirical validation of the results. The AC agrees.

**Justification For Why Not Higher Score:**

The paper will benefit from another round of revision, due to a limited empirical validation of the method!

**Justification For Why Not Lower Score:**

N/A

---

### Decision · Program_Chairs · 2024-01-16

Reject